# Structure of a VirD4 coupling protein bound to a VirB type IV secretion machinery

Adam Redzej[1,†], Marta Ukleja[1,†], Sarah Connery[1,†], Martina Trokter[1], Catarina Felisberto-Rodrigues[1], Adam Cryar[2], Konstantinos Thalassinos[1,2], Richard D Hayward[1,2], Elena V Orlova[1,*] iD &
Gabriel Waksman[1,2,**] iD

## Abstract

Type IV secretion (T4S) systems are versatile bacterial secretion systems mediating transport of protein and/or DNA. T4S systems are generally composed of 11 VirB proteins and 1 VirD protein (VirD4). The VirB1-11 proteins assemble to form a secretion machinery and a pilus while the VirD4 protein is responsible for substrate recruitment. The structure of VirD4 in isolation is known; however, its structure bound to the VirB1-11 apparatus has not been determined. Here, we purify a T4S system with VirD4 bound, define the biochemical requirements for complex formation and describe the protein–protein interaction network in which VirD4 is involved. We also solve the structure of this complex by negative stain electron microscopy, demonstrating that two copies of VirD4 dimers locate on both sides of the apparatus, in between the VirB4 ATPases. Given the central role of VirD4 in type IV secretion, our study provides mechanistic insights on a process that mediates the dangerous spread of antibiotic resistance genes among bacterial populations.

**Keywords** bacterial conjugation; structure; type 4 secretion system; VirD4
**Subject Categories** Microbiology, Virology & Host Pathogen Interaction; Structural Biology
The EMBO Journal (2017) 36: 3080–3095

## Introduction

Type IV secretion (T4S) systems are membrane embedded multiprotein complexes, which are present, both in Gram-negative and Gram-positive bacteria, as well as archaea (Wallden *et al*, 2010; Trokter *et al*, 2014; Costa *et al*, 2015). They are used to transport a variety of biomolecules across the bacterial envelope. Depending on the type of transported substrates, they can be divided into three groups (Alvarez-Martinez & Christie, 2009). The first group of T4S

systems are called conjugative T4S systems and members of this group transfer DNA from a donor to a recipient cell (de la Cruz *et al*, 2010). This process contributes to the spread of antibiotic resistance genes among different bacterial species and is also instrumental in bacterial adaptation to environmental changes (Thomas & Nielsen, 2005). The second group of T4S systems are responsible for DNA release or uptake to or from the extracellular milieu, respectively: these T4S systems operate in bacterial species such as *Neisseria gonorrhoeae* and *Helicobacter pylori* (Karnholz *et al*, 2006; Ramsey *et al*, 2011). Finally, T4S systems can translocate effector molecules into eukaryotic cells, which can result in many diseases such as Legionnaires' disease caused by *Legionella pneumophila* or brucellosis caused by *Brucella* spp (Llosa *et al*, 2009; Nagai & Kubori, 2011; Terradot & Waksman, 2011).

The conjugative T4S systems are generally composed of 12 proteins, named after the T4S system from *Agrobacterium tumefaciens* responsible for transfer of the T-DNA of the Ti plasmid into plants. In this system, T4S system components are termed VirB1-VirB11 and VirD4 (Fronzes *et al*, 2009a). The central channel of the T4S system, called the "core" complex, is formed from 14 copies of the proteins VirB7, VirB9 and VirB10, and spans both membranes (Fronzes *et al*, 2009b; Rivera-Calzada *et al*, 2013). The overall structure of the core complex appears as a two-layered ring with 14-fold symmetry. The outer layer (O-layer) is made of full-length VirB7 and the C-terminal domains of VirB9 and VirB10 and is anchored in the outer membrane via lipidated VirB7 and an α-helical barrel (Chandran *et al*, 2009). The inner-layer (I-layer) is assembled from the N-terminal parts of VirB9 and VirB10 with the very N-terminal part of VirB10 extending into the cytosol and inserting a transmembrane helix into the inner membrane. Recently, the structure of a larger complex containing eight VirB components including the VirB3-10 proteins (a complex referred to thereafter as T4SS$_{3-10}$) was solved by negative stain electron microscopy (NS-EM), revealing the structure of the inner membrane part of a T4S system or inner membrane complex (IMC) composed of VirB3-6 and VirB8 (Low *et al*, 2014). The most prominent features of the IMC are two

1 Department of Biological Sciences, Institute of Structural and Molecular Biology, Birkbeck, London, UK
2 Division of Biosciences, Institute of Structural and Molecular Biology, University College of London, London, UK
 *Corresponding author. Tel: +44 2076316833; E-mail: e.orlova@mail.cryst.bbk.ac.uk
 **Corresponding author. Tel: +44 2076316845; E-mail: g.waksman@mail.cryst.bbk.ac.uk
 †These authors contributed equally to this work
 [The copyright line of this article was changed on 25 September 2017 after original online publication.]

barrel-like structures which are each divided into three layers: the lower, middle and upper tiers. These barrels are connected by an arch region to a stalk region, which links the IMC to the core complex. While each of the barrels is comprised of six copies of VirB4, the accurate location of VirB3, VirB5, VirB6 and VirB8 within the IMC is not clear. Extending from the core complex outside of the cell is an appendage called a pilus, which is a polymer of VirB2 and is essential for making the connection between the recipient and donor cells. Recent high-resolution cryo EM structure of the pilus isolated from the F-like plasmid shows that the inside of the pilus is covered with lipid head groups, which might facilitate the transfer of the DNA between cells (Costa *et al*, 2016; Hospenthal *et al*, 2017).

The energy powering the T4S system is supplied by three ATPases VirB4, VirB11 and VirD4, which are located on the cytoplasmic side of the T4S system. While VirB4 together with VirB11 are required for pilus biogenesis and substrate translocation, VirD4 plays the role of a coupling protein, responsible for the recruitment of the substrate to the T4S system channel (Cabezon *et al*, 1997, 2015; Cascales & Christie, 2004). VirB11 has been reported as a hexamer in different systems (Machon *et al*, 2002; Savvides *et al*, 2003; Hare *et al*, 2006). VirD4 and VirB4, though expected to form hexamers, have been identified in monomeric and dimeric forms as well (Gomis-Ruth *et al*, 2001; Schroder *et al*, 2002; Rabel *et al*, 2003; Arechaga *et al*, 2008; Mihajlovic *et al*, 2009; Durand *et al*, 2011; Pena *et al*, 2012; Wallden *et al*, 2012). All three ATPases interact with one another and these interactions likely direct the two processes in which T4S systems are involved: pilus biogenesis and substrate secretion (Atmakuri *et al*, 2004; Ripoll-Rozada *et al*, 2013). VirD4 also interacts with the N-terminally located transmembrane helices of VirB10 (Llosa *et al*, 2003; Ripoll-Rozada *et al*, 2013; Segura *et al*, 2013). Although the general organization of the conjugative T4S system is well known, the details of the substrate secretion mechanism are still unclear. Most of the knowledge about the path of the secreted DNA comes from the *A. tumefaciens* system where the proteins with which the DNA interacts on its way through the secretion channel have been identified. During transfer, a single strand of DNA makes successive contact with the VirD4 coupling protein, and then the VirB11 ATPase, the inner membrane proteins VirB6 and VirB8 and finally the core complex proteins VirB7 and VirB9 (Cascales & Christie, 2004).

Conjugative transfer of plasmid DNA requires the processing of the plasmid DNA by a large soluble complex containing 3–4 proteins, the largest of which is called the relaxase (Ilangovan *et al*, 2017). The relaxase is loaded onto a region of the plasmid called the origin of transfer or *oriT* with the assistance of 2–3 auxiliary proteins (Zechner *et al*, 2012; Ilangovan *et al*, 2015). Once loaded, the relaxase nicks *oriT* at a site called *nic*, and covalently reacts to the resulting 5′-phosphate. It is this covalent relaxase–ssDNA complex that is then recruited to the VirD4 coupling protein for transport through the T4S system.

The overall goal of the study presented here is to investigate the structure of the coupling protein VirD4 in the context of the entire T4S system encoded by the R388 conjugative plasmid. We purify a complex containing the eight VirB3-10 proteins (named TrwM-TrwE in the R388 nomenclature) to which the VirD4 protein (TrwB) is bound (a complex termed "T4SS$_{3\text{-}10+D4}$"), determine the stoichiometry of TrwB$_{/VirD4}$ within this complex, define the biochemical requirements for complex formation by systematically deleting each one of the T4S system IMC component proteins and evaluating their impact on complex formation and characterize the protein–protein interaction network of TrwB$_{/VirD4}$ with T4S system components within the machinery. Finally, we present the NS-EM structure of the T4SS$_{3\text{-}10+D4}$ complex, which identifies extra density corresponding to TrwB$_{/VirD4}$ within the T4S machinery, and confirm both presence and location of TrwB$_{/VirD4}$ using cross-linking mass spectrometry (XL-MS) and immuno-labelling followed by NS-EM. This study reveals a remarkable IMC where each multimeric ATPase, TrwK$_{/VirB4}$ and TrwB$_{/VirD4}$, is present in two copies and is located at the nexus of the T4S system where they can couple protein and DNA transport in a remarkably efficient manner.

# Results and Discussion

### Expression of functional T4S systems

R388 is a conjugative plasmid approximately 30 kb in size, which constitutively expresses the T4S system and relaxosome genes (Bolland *et al*, 1990; Llosa *et al*, 1994). In this conjugative system, the *virB1-11* genes are called *trwN-D* while the genes encoding the VirD4 coupling protein and the relaxase are called *trwB* and *trwC*, respectively. Relaxosome formation requires two auxiliary proteins, plasmid-encoded TrwA, and host-encoded IHF. *trwN-D* and *trwABC* are located in two operons, the expression of which is coordinated by three distinct promoters (de Paz *et al*, 2005; Fernandez-Lopez *et al*, 2006). For clarity, we will use both the "Trw" and "VirB" nomenclatures.

For the purpose of this study, three requirements for cloning were considered key: (i) sufficient material for NS-EM analysis should be obtained, (ii) tags should be easily engineered within the vectors employed, and (iii) all combinations of constructs should drive the production of functional T4S systems. All constructs generated in this study are listed in Appendix Table S1.

First, the *trwN$_{/virB1}$-trwD$_{/virB11}$* operon was cloned into the vector pBADM11 (referred to as pBADM11_*trwN$_{/virB1}$-trwD$_{/virB11}$*), which is under the regulation of the inducible *ara* promoter. In parallel, the *trwABC* genes encoding for the relaxosome components as well as the coupling protein were cloned into the pCDFduet vector (referred to as pCDF_*trwABC*). A third construct was generated containing the R388 plasmid *oriT* sequence in the pRSF plasmid (referred to as pRSF_*oriT*). All plasmids are compatible, having a different origin of replication. As it is crucial to ascertain that expressed T4S system and relaxosome components are assembled properly, bacteria containing the combination of three plasmids (the pBADM11_ *trwN$_{/virB1}$-trwD$_{/virB11}$* under arabinose control, the *trwABC*-containing plasmid under IPTG control and the *oriT*-containing plasmid) were tested for their ability to conjugate the *oriT* –containing plasmid into recipient cells. Conjugation experiments were conducted, and conjugation efficiencies were determined as described in Materials and Methods. As reported in Table 1, the three-plasmid system efficiently mediates conjugative transfer. These results are also consistent with the findings by Bolland *et al* (1990) that the *trw* genes can be taken out of their R388 plasmid context to mediate conjugation.

For purification of T4S system complexes, the pBADM11_ *trwN$_{/virB1}$-trwD$_{/virB11}$* expression clone was modified. Previous studies on T4S system have exploited a Strep-tag located at the

**Table 1. Conjugation efficiencies of various R388 T4SS constructs.**

| T4S plasmid | Relaxosome plasmid | Mating efficiency |
|---|---|---|
| pBADM11_$trwN_{/virB1}$-$trwD_{/virB11}$ | pCDF_$trwABC$ | 0.5 |
| pBADM11_$trwN_{/virB1}$-$trwD_{/virB11}$ | pCDF_$trwAC$ | $< 10^{-6}$ |
| pBADM11_$trwN_{/virB1}$-$trwE_{/virB10Strep\_}$$trwD_{/virB11\_His}trwB_{/virD4}$ | pCDF_$trwAC$ | 0.07 |
| pBADM11_ $trwB_{/virD4His}trwN_{/virB1}$-$trwE_{/virB10Strep\_}trwD_{/virB11}$ | pCDF_$trwAC$ | 0.05 |

The mating efficiencies obtained with plasmids expressing different variations of the T4S system genes from R388 are reported.

C-terminus of VirB10 (Chandran et al, 2009; Fronzes et al, 2009b; Wallden et al, 2012; Rivera-Calzada et al, 2013; Low et al, 2014). Thus, a sequence encoding a Strep-tag was introduced at this position of the $trwE_{/virB10}$ gene. The resulting clone is named pBADM11_$trwN_{/virB1}$-$trwE_{/virB10Strep\_}trwD_{/virB11}$.

Using this construct, two additional clones were produced: i—pBADM11_$trwN_{/virB1}$-$trwE_{/virB10Strep\_}trwD_{/virB11\_His}trwB_{/virD4}$, with the $TrwB_{/VirD4}$-encoding gene located after the sequence encoding $TrwD_{/VirB11}$ and N-terminally His-tagged, and ii—pBADM11_$trwB_{/virD4His}trwN_{/virB1}$-$trwE_{/virB10Strep\_}trwD_{/virB11}$, with the $TrwB_{/VirD4}$-encoding gene inserted before the sequence encoding $TrwN_{/VirB1}$ and C-terminally His-tagged. To test whether these constructs induce formation of functional T4S systems, we generated a variant of pCDF_$trwABC$ where the $trwB_{/virD4}$ gene was deleted (resulting in a construct named pCDF_$trwAC$). The pBADM11_$trwN_{/virB1}$-$trwE_{/virB10Strep\_}$$trwD_{/virB11\_His}trwB_{/virD4}$ or pBADM11_$trwB_{/virD4His}trwN_{/virB1}$-$trwE_{/virB10Strep\_}trwD_{/virB11}$ constructs were then assayed using pCDF_$trwAC$ and pRSF_$oriT$ for their ability to mediate conjugation of pRSF_$oriT$. As shown in Table 1, conjugation is still observed but reduced by approximately 10-fold in both cases, indicating that these constructs still assemble functional T4S system and relaxosome complexes, albeit at a reduced efficiency compared to wild type. Conjugation does not occur when $trwB_{/virD4}$ is removed.

## VirD4 binds to the T4SS$_{3-10}$ complex

Having obtained functional conjugative T4S machineries, we next asked whether any of these machineries could be extracted from the bacterial envelope and purified. We therefore induced expression of the pBADM11_$trwN_{/virB1}$-$trwE_{/virB10Strep\_}trwD_{/virB11\_His}trwB_{/virD4}$ or the pBADM11_$trwB_{/virD4His}trwN_{/virB1}$-$trwE_{/virB10Strep\_}trwD_{/virB11}$ constructs and submitted the detergent-solubilized membrane extracts to two consecutive tag-affinity chromatography steps, first using the Strep-tag on $TrwE_{/VirB10}$ and then the His tag on $TrwB_{/VirD4}$. The resulting materials were analysed by SDS–PAGE. Both constructs yielded similar results, and therefore, all subsequent experiments were carried out using the pBADM11_$trwN_{/virB1}$-$trwE_{/virB10Strep\_}trwD_{/virB11\_His}trwB_{/virD4}$ construct. We observed 10 bands (Fig 1A, left panel). These were identified by mass spectrometry to be (from higher to lower molecular weight) $TrwK_{/VirB4}$, $TrwB_{/VirD4}$, $TrwE_{/VirB10}$, $TrwG-F_{/VirB8-9}$, $TrwI_{/VirB6}$, $TrwJ_{/VirB5}$, $TrwM_{/VirB3}$ and $TrwH_{/VirB7}$. The presence of $TrwB_{/VirD4}$ was confirmed by purifying the T4SS$_{3-10}$ complex from cells expressing the pBADM11_$trwN_{/virB1}$-$trwE_{/virB10Strep\_}trwD_{/virB11}$ construct. In the SDS–PAGE profile of the complex purified from $TrwB_{/VirD4}$-producing cells, a clear

band of a size between 50 and 60 kDa is observed, which is absent in the T4SS$_{3-10}$ complex (Fig 1A, right panel). The theoretical molecular weight of $TrwB_{/VirD4}$ incorporating 10 additional Histidine residues is 57 kDa. The Western blot analysis using $TrwB_{/VirD4}$ antibodies (this study; see Materials and Methods for $TrwB_{/VirD4}$ antibody production) confirmed that this band corresponds to the $TrwB_{/VirD4}$ protein (Fig EV1A). All other bands observed in the SDS–PAGE analysis of the complex purified from $TrwB_{/VirD4}$-producing cells are also present in the T4SS$_{3-10}$ complex. This includes OmpC and OmpA, contaminants previously proven to be minor (Low et al, 2014). Thus, the complex purified from $TrwB_{/VirD4}$-producing cells is the T4SS$_{3-10}$ complex to which VirD4 is bound. We term this complex the T4SS$_{3-10+D4}$ complex.

We next determined the stoichiometry of $TrwB_{/VirD4}$ within the T4SS$_{3-10+D4}$ complex using cysteine labelling (Fig 1B). Free thiol groups within the proteins of the complex were reacted using Alexa Fluor633 coupled to maleimide. The labelled sample of the complex was analysed using SDS–PAGE, and the fluorescence intensity of the bands was quantified. The $TrwE_{/VirB10}$ protein as a part of the $TrwH_{/VirB7}$-$TrwF_{/VirB9}$-$TrwE_{/VirB10}$ core complex alone or of the T4SS$_{3-10}$ complex is known to be present in 14 copies (Fronzes et al, 2009b; Low et al, 2014). Therefore, the fluorescence signal quantified from the $TrwE_{/VirB10}$ band was used as a reference to determine the stoichiometry of the other components. As a result, the amount of $TrwB_{/VirD4}$ within the sample can be derived as $4.5 \pm 0.2$ $TrwB_{/VirD4}$ subunits per complex (calculated from three independent preparations of the T4SS$_{3-10+D4}$ complex; Fig 1B). All other components in the T4SS$_{3-10+D4}$ complex were present in the same stoichiometry observed previously in the T4SS$_{3-10}$ complex.

## Systematic deletions of single T4SS IMC components reveal different subcomplexes of the T4SS$_{3-10+D4}$ complex

To better understand how the absence of different components of the T4SS affect T4SS$_{3-10+D4}$ complex formation, single $trw_{/virB}$ gene deletion mutants within the pBADM11_$trwN_{/virB1}$-$trwE_{/virB10Strep\_}$$trwD_{/virB11\_His}trwB_{/virD4}$ construct were generated. These constructs will be referred to as $\Delta trwX_{/virBY}$ deletion constructs, X (a letter) or Y (a number) identifying the component gene deleted from the cluster using the Trw or VirB nomenclature, respectively (Appendix Table S1). Only deletions of IMC components ($TrwL_{/VirB2}$, $TrwM_{/VirB3}$, $TrwK_{/VirB4}$, $TrwJ_{/VirB5}$, $TrwI_{/VirB6}$, $TrwG_{/VirB8}$ and $TrwD_{/VirB11}$) were investigated as core OMC components, $TrwH_{/VirB7}$, $TrwF_{/VirB9}$ and $TrwE_{/VirB10}$, are already known to be indispensable for complex assembly (Fronzes et al, 2009b), and $TrwN_{/VirB1}$ is not an essential part of the T4S system (Berger & Christie, 1994). These constructs were then used to produce and purify subcomplexes using the same purification protocol employed for the purification of the T4SS$_{3-10+D4}$ complex, that is a pull-down using the Strep-tag on $TrwE_{/VirB10}$ followed by a pull-down using the His tag on $TrwB_{/VirD4}$. Resulting complexes were analysed using SDS–PAGE to compare which proteins are still present within the complex in the absence of individual T4S system component (Fig 1C). Three distinct complexes could be observed. In the absence of the $TrwL_{/VirB2}$, $TrwJ_{/VirB5}$ or $TrwD_{/VirB11}$, all of the other components of the T4SS$_{3-10+D4}$ complex are still present. Therefore, none of the pilus subunits or $TrwD_{/VirB11}$ are essential for the stability of the T4SS$_{3-10+D4}$ complex. However, deletion of $TrwM_{/VirB3}$ or

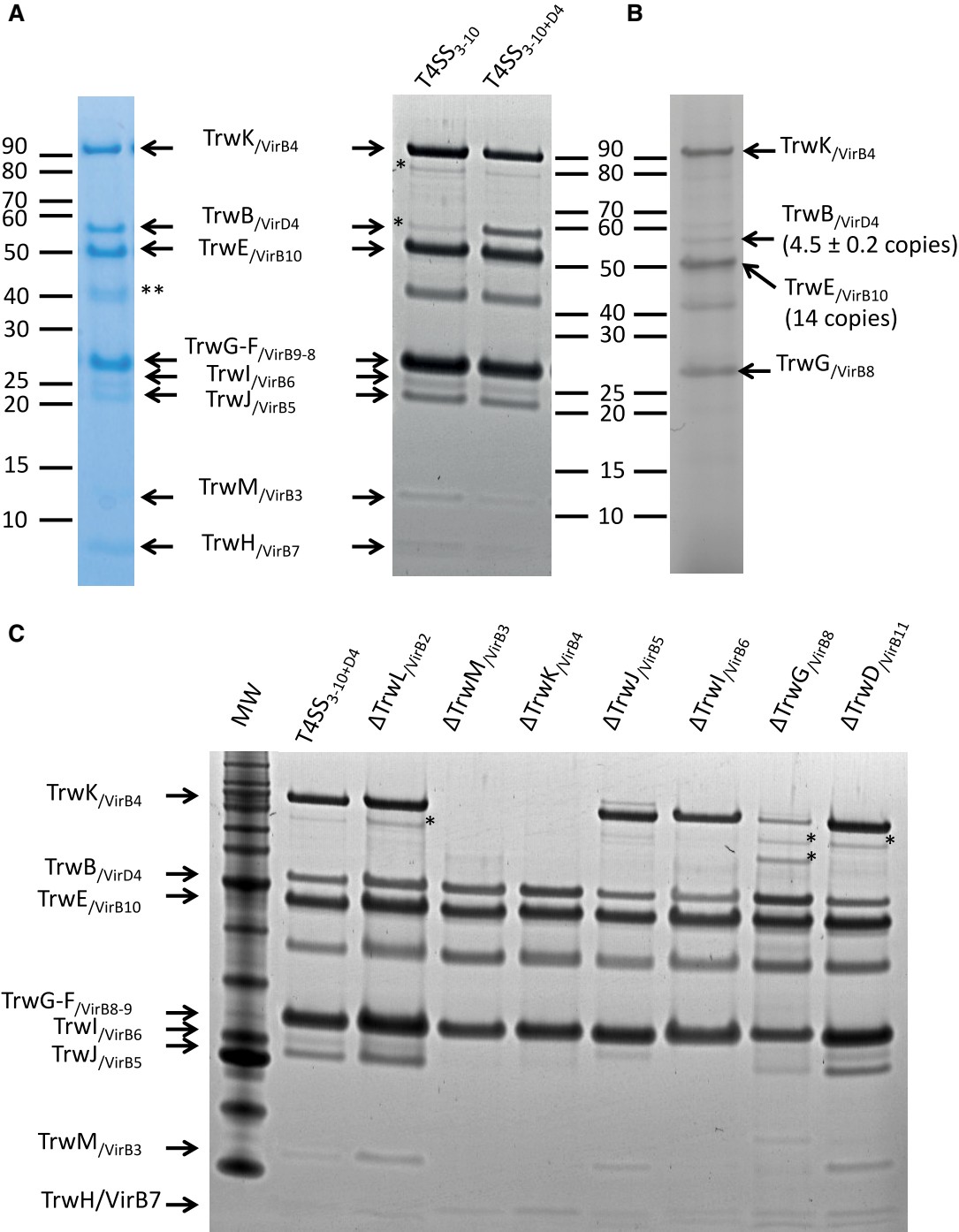

**Figure 1.  SDS–PAGE analysis of purified T4SS$_{3-10+D4}$ complex, comparison with the T4SS$_{3-10}$ complex, stoichiometry measurement and purification of T4SS$_{3-10+D4}$ subcomplexes after deletion of single T4S system components.**

A  SDS–PAGE analysis of purified T4SS$_{3-10+D4}$ complex. Left panel: Coomassie-stained SDS–PAGE gel of the T4SS$_{3-10+D4}$ complex. Right panel: Sypro Ruby-stained SDS–PAGE gel of the T4SS$_{3-10m}$ and T4SS$_{3-10+D4}$ complexes. Molecular weights of MW markers are shown on the left. Identification of bands is shown in the middle. *Indicates TrwK$_{/VirB4}$ degradation products. **Indicates minor contaminants OmpA and OmpC.

B  Stoichiometry measurement. Fluorescence scan of an SDS–PAGE gel analysis of the T4SS$_{3-10+D4}$ complex where cysteine residues were reacted with Alexa Fluor 633 C$_5$ maleimide. Signals corresponding to the proteins coupled to Alexa Fluor 633 were detected at a wavelength of 633 nm. Fluorescent Trw protein bands are labelled, and the derived stoichiometry for TrwB$_{/VirD4}$ is indicated.

C  SDS–PAGE analysis of complexes purified from cell expressing deletion mutants of the pBADM11_*trwN$_{/virB1}$-trwE$_{/virB10Strep}$_trwD$_{/virB11\_His}$trwB$_{/virD4}$* constructs termed pBADM11_*ΔtrwX$_{/ΔvirBY}$* (see main text). The same molecular weight marker has been used on this gel as in the gel presented in panel (A). *Indicates degradation products of TrwK$_{/VirB4}$.

Source data are available online for this figure.

TrwK$_{/VirB4}$ results in the purification of a complex composed of the core OMC components, TrwH$_{/VirB7}$, TrwF$_{/VirB9}$ and TrwE$_{/VirB10}$, as well as TrwB$_{/VirD4}$, indicating that TrwB$_{/VirD4}$ interacts directly with the core complex, presumably through interaction with the N-terminus of TrwE$_{/VirB10}$ (Llosa *et al*, 2003; Segura *et al*, 2013). Similarly, when pulling the complex using the Δ*trwG$_{/virB8}$* construct, apart from TrwH$_{/VirB7}$, TrwF$_{/VirB9}$, TrwE$_{/VirB10}$ and TrwB$_{/VirD4}$, only traces of TrwK$_{/VirB4}$ and its degradation products can be detected, indicating that TrwG$_{/VirB8}$ serves to stabilize TrwK$_{/VirB4}$ within the IMC. Interestingly, when the *trwI$_{/virB6}$* gene is not expressed with the other T4SS components, TrwJ$_{/VirB5}$ is no longer present in the complex. This might imply an interaction between these two components within the T4SS$_{3-10+D4}$ complex and is consistent with the known effect of VirB6 on stabilizing VirB5 in *A. tumefaciens* systems (Hapfelmeier *et al*, 2000). TrwD$_{/VirB11}$ appears to play no role in T4SS$_{3-10+D4}$ assembly as its deletion has no significant impact on the composition of the complex purified.

These conclusions are only valid if gene deletion does not affect expression of other genes. Although Larrea *et al* (2013) have already shown using gene complementation that individual *trw/virB* gene disruptions do not show any polar effect and that complementation *in trans* results in functional conjugative systems (except for a disruption of *trwL$_{/virB2}$* that was shown to have a polar effect on the expression of *trwK$_{/virB4}$*), we set out to show biochemically that deletions of single Trw$_{/VirB}$ proteins do not affect production and stability of others. Thus, for each deletion constructs for which an effect on complex formation was observed (Δ*trwM$_{/virB3}$*, Δ*trwK$_{/virB4}$*, Δ*trwI$_{/virB6}$* and Δ*trwG$_{/virB8}$*), we introduced sequences encoding

FLAG tags to the 5′- or 3′-ends of each component gene that we observed were missing after pull-down, and monitored production of the corresponding proteins. Since, in the Δ*trwM$_{/virB3}$* deletion, we observed the loss of TrwK$_{/VirB4}$, TrwJ$_{/VirB5}$, TrwI$_{/VirB6}$ and TrwG$_{/VirB8}$, four derivatives of this Δ*trwM$_{/virB3}$* construct were generated in order to produce variants of these four proteins containing a FLAG tag at either their N- or C-terminus (Appendix Table S1). Similarly, the Δ*trwK$_{/virB4}$* deletion results in the loss of the TrwM$_{/VirB3}$, TrwJ$_{/VirB5}$, TrwI$_{/VirB6}$ and TrwG$_{/VirB8}$ proteins: thus, we generated four further variants of the Δ*trwK$_{/virB4}$* construct where these proteins were FLAG-tagged individually (Appendix Table S1). For the Δ*trwI$_{/virB6}$* deletion, we observed the loss of TrwJ$_{/VirB5}$, and consequently, a FLAG sequence was added to TrwJ$_{/VirB5}$ and its production checked (Appendix Table S1). Finally, deletion of TrwG$_{/VirB8}$ resulted in the destabilization of TrwM$_{/VirB3}$, TrwK$_{/VirB4}$, TrwJ$_{/VirB5}$ and TrwI$_{/VirB6}$, and therefore, four additional constructs of this deletion mutant were generated to include a FLAG sequence in each of these proteins, one at a time (Appendix Table S1). To compare production levels of each of the FLAG-tagged proteins within deletion mutants with the amounts produced in the non-deleted wild-type pBADM11_*trwN$_{/virB1}$-trwE$_{/virB10Strep}$_trwD$_{/virB11-}$*$_{His}$*trwB$_{/virD4}$* strain, FLAG sequences were also inserted in the wild-type construct at the same position as in the deletion mutants, one at a time (Appendix Table S1). After induction, expression was monitored by Western blot analysis using an anti-FLAG antibody. As can be seen in Fig EV1B, all proteins are produced. Lower expression compared to wild type is however observed for TrwM$_{/VirB3}$ in the Δ*trwK$_{/virB4}$* construct and for TrwI$_{/VirB6}$ in the

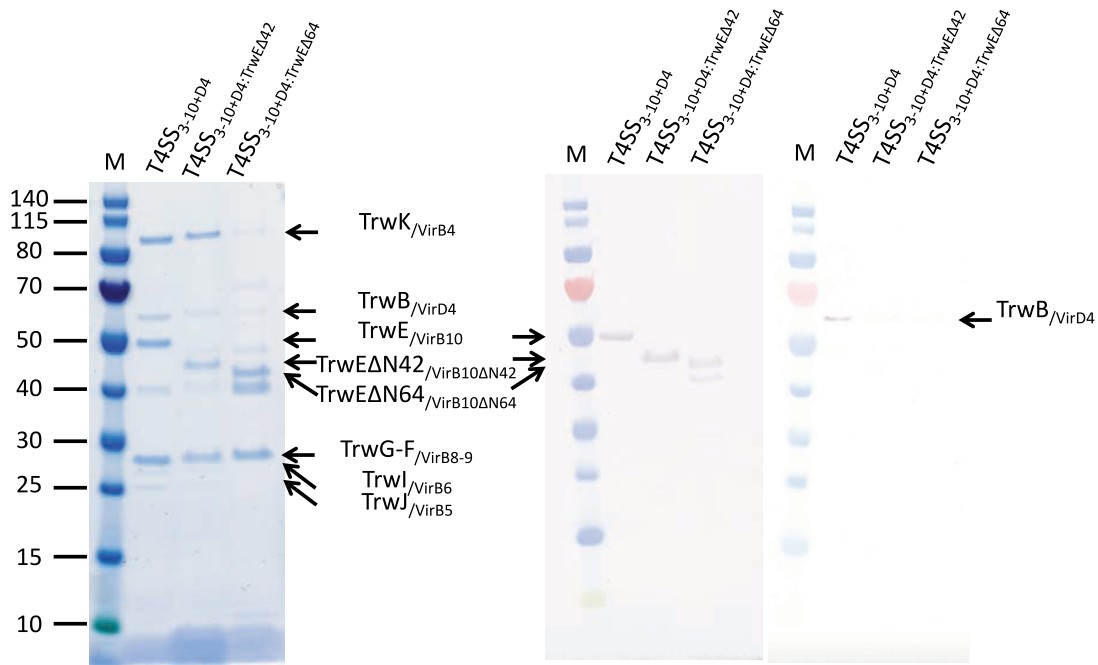

**Figure 2.  Interaction between TrwB$_{/VirD4}$ and TrwE$_{/VirB10}$.**

Left panel: Coomassie-stained SDS–PAGE gel of the T4SS$_{3-10+D4}$ complex (lane 1), T4SS$_{3-10+D4:TrwEΔN42}$ complex (lane 2) and T4SS$_{3-10+D4:TrwEΔN64}$ complex (lane 3). Middle panel: Western blot analysis using αStrep antibodies to detect the TrwE$_{/VirB10}$ protein within the T4SS$_{3-10+D4}$ complex and its variants. Right panel: Western blot analysis using αHis antibodies to detect the TrwB$_{/VirD4}$ protein within the T4SS$_{3-10+D4}$ complex and its variants.

Source data are available online for this figure.

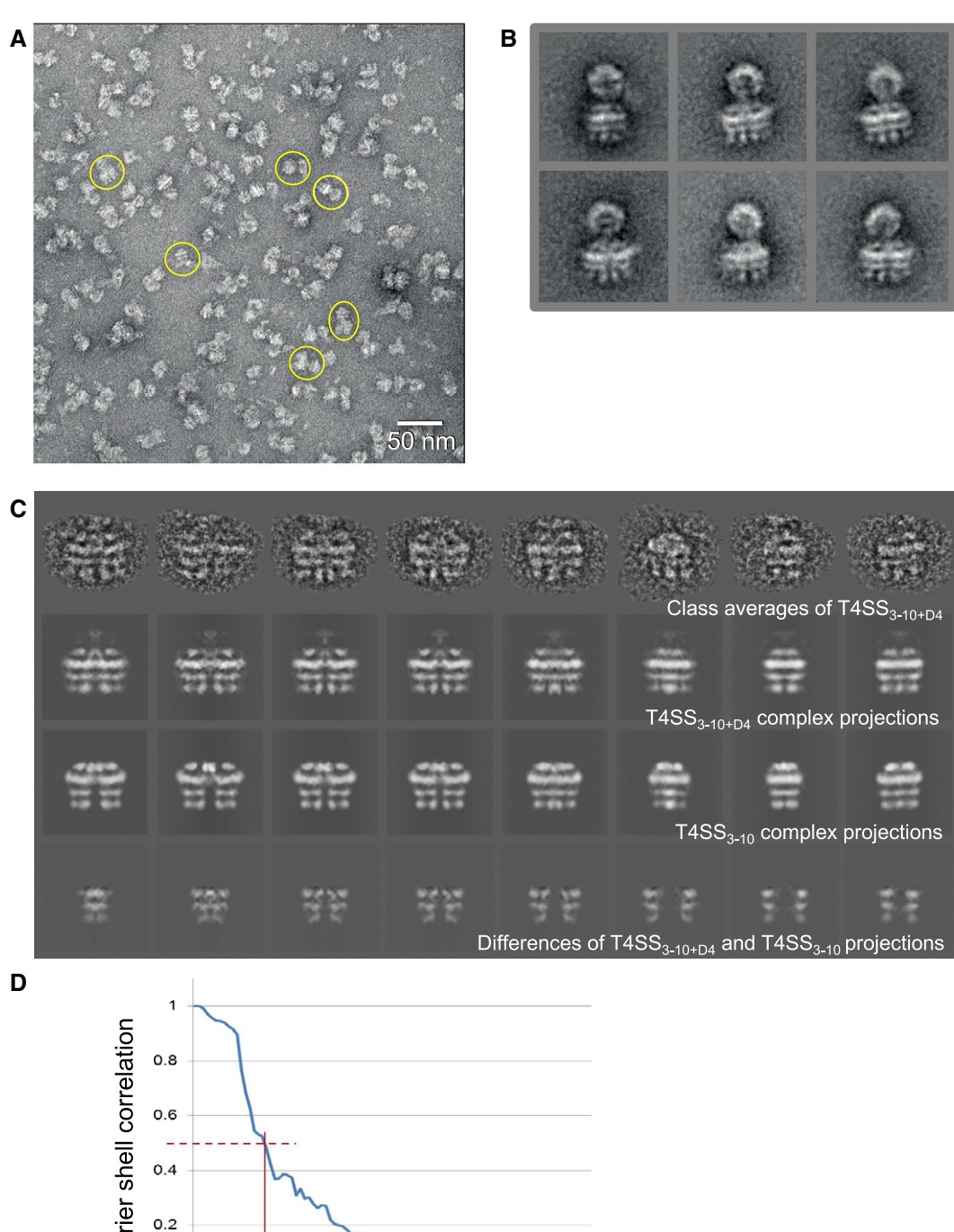

**Figure 3.  NS-EM of the T4SS$_{3-10+D4}$ complex.**

A    NS-EM micrograph of the T4SS$_{3-10+D4}$ complex. A few representative particles of the T4SS$_{3-10+D4}$ complex are circled in yellow. Scale bar 50 nm.

B    Representative class averages of T4SS$_{3-10+D4}$ complex obtained after initial alignment and classification.

C    The upper row represents typical class averages of the T4SS$_{3-10+D4}$ IMC complex (i.e. with the outer membrane core complex (OMC) masked out). The second row shows projections of the T4SS$_{3-10+D4}$ IMC complex in the same directions determined for the classes above. The third row displays projections of the T4SS$_{3-10}$ IMC complex (EMD-2567) in the same directions. The bottom row shows the differences between the projections above (plus or minus TrwB$_{NVirD4}$) corresponding to positions of the TrwB$_{NVirD4}$ protein.

D    FSC of the electron density map of the T4SS$_{3-10+D4}$ complex. The 0.5 line crosses the FSC at a resolution of 28 Å.

Source data are available online for this figure.

$\Delta trwM_{/virB3}$ and $\Delta trwK_{/virB4}$ constructs. Lower production of TrwM$_{/VirB3}$ and TrwI$_{/VirB6}$ in these constructs might be due to an effect of the removed protein on the stability of the TrwM$_{/VirB3}$ or TrwI$_{/VirB6}$ proteins. Indeed, VirB3, 4, 6 are known to interact and stabilize each other (Hapfelmeier *et al*, 2000; Mossey *et al*, 2010). Nevertheless, TrwM$_{/VirB3}$ and TrwI$_{/VirB6}$ are produced in the $\Delta trwM_{/virB4}$ and $\Delta trwK_{/virB3}$ constructs, respectively, and, although the level of expression is lower than in the corresponding wild-type constructs, it is similar to that observed for TrwJ$_{/VirB5}$ in all constructs including wild type, and thus cannot be considered limiting. We also checked for expression of $trwB_{/virD4}$ in all deletion clusters and showed that each produced similar quantities of TrwB$_{/VirD4}$ (Fig EV1A). We therefore conclude that the loss of particular T4SS components during pull-down that we observe when some T4SS components are removed is not due to the lack of expression of these particular components. We however note that some of these expression results contradict those reported in Larrea *et al* (2013): for example, they found that production of TrwJ$_{/VirB5}$ was abolished in all insertion mutants, while, in our study, TrwJ$_{/VirB5}$ is produced in equal measure in all deletion mutants and in the wild-type construct.

This investigation of complex formation of TrwB$_{/VirD4}$ with the T4SS$_{3-10}$ complex pointed to the importance of the interaction between TrwB$_{/VirD4}$ with the core TrwH$_{/VirB7}$-TrwF$_{/VirB9}$-TrwE$_{/VirB10}$ complex. Interaction between TrwB$_{/VirD4}$ and TrwE$_{/VirB10}$ has been previously observed, and therefore, we set out to confirm it biochemically in the context of the entire T4SS$_{3-10+D4}$ complex. Within the pBADM11_$trwN_{/virB1}$-$trwE_{/virB10Strep}$_$trwD_{/virB11-}$ $_{His}trwB_{/virD4}$ cluster, we deleted sequences encoding either the entire cytoplasmic tail of TrwE$_{/VirB10}$ (residues 1–42; referred to as T4SS$_{3-10+D4:TrwE\Delta42}$) or both the cytoplasmic and transmembrane segment of this protein (residues 1–64; referred to as T4SS$_{3-10+D4:}$ $_{TrwE\Delta64}$). After induction of expression, the T4SS$_{3-10+D4}$ complex and variants were purified and analysed by SDS–PAGE and Western blot (Fig 2). Expression of the T4SS$_{3-10+D4:TrwE\Delta42}$ resulted in lower amounts of TrwB$_{/VirD4}$ bound to the complex indicating that indeed the cytoplasmic N-terminal region of TrwE$_{/VirB10}$ is important for interaction with TrwB$_{/VirD4}$. Removal of the transmembrane segment of TrwE$_{/VirB10}$ further destabilize the interaction with the coupling protein, confirming that the interactions between TrwE$_{/VirB10}$ and TrwB$_{/VirD4}$ involve the transmembrane segment of both proteins.

## Negative stain electron microscopy of the T4SS$_{3-10+D4}$ complex enables localization of the TrwB$_{/VirD4}$ coupling protein within the complex

To locate TrwB$_{/VirD4}$ within the T4SS$_{3-10+D4}$ complex, we solved the structure of this complex using NS-EM and compared it to that of the T4SS$_{3-10}$ solved previously (Low *et al*, 2014) using the same method (Figs 3 and 4). Since TrwD$_{/VirB11}$ is not part of the T4SS$_{3-10+D4}$ complex, we produced the complex using a construct where the $trwD_{/virB11}$ gene was removed (construct $\Delta trwD_{/virB11}$ (see above)). Moreover, in order to prevent TrwB$_{/VirD4}$ from dissociating during sample and grid preparation, the complex was cross-linked and re-purified using the GraFix method prior to the NS-EM analysis (Stark, 2010). As for the T4SS$_{3-10}$ complex, images of single particles (Fig 3A) and class averages (Fig 3B) of the T4SS$_{3-10+D4}$ clearly show

that the complex is made of two distinct parts, the core OMC and the IMC, which are connected by the less defined density of the stalk region (Fig 3B and C, upper row). A certain degree of flexibility can also be observed in the stalk region. As TrwB$_{/VirD4}$ is an inner membrane protein, the image processing focused on the reconstruction of the IMC of the T4SS$_{3-10+D4}$ complex. This was achieved to a resolution of 28 Å (Fig 3D). Similarly to the T4SS$_{3-10}$ IMC, two large barrel-like densities are observed (in yellow in Fig 4). These can be clearly divided into three layers, upper, middle and lower (Fig 4A). They superimpose well with the two barrel-like structures identified in the previously determined T4SS$_{3-10}$ complex structure as formed by TrwK$_{/VirB4}$ trimers of dimers (Fig 4A; compare left and right panels). Density connects the two barrels on the top, a region previously characterized as "the arches" (in green in Fig 4A–C). The most significant differences between the structures of the T4SS$_{3-10}$ and T4SS$_{3-10+D4}$ IMCs are two extra regions of density present between the two barrel-like TrwK$_{/VirB4}$ structures on both sides of the complex (Fig 3C, 2$^{nd}$, 3$^{rd}$ and 4$^{th}$ rows; in blue in Fig 4A, right panel; and Fig EV2A and B). Since the major difference in composition between the two complexes is the presence of TrwB$_{/VirD4}$ in the T4SS$_{3-10+D4}$ complex, we conclude that this density must correspond to TrwB$_{/VirD4}$. Importantly, its shape has similarities to that of TrwK$_{/VirB4}$ in that it is also formed of three distinct layers. TrwB$_{/VirD4}$ and TrwK$_{/VirB4}$ have been shown to have very similar structures in spite of low sequence similarities (Wallden *et al*, 2012). Thus, it is not surprising that they should share similar density features (Fig 4A). Interestingly, each density is not large enough to accommodate a hexamer of TrwB$_{/VirD4}$. Instead, only two TrwB$_{/VirD4}$ molecules appear to fit (Fig EV2C). However, because negative stain is known to cause flattening and distortion, an accurate evaluation of the number of molecules fitting these regions cannot be obtained. Nevertheless, the fitting of two molecules is consistent with the stoichiometry measurement, and therefore, it is likely that only two TrwB$_{/VirD4}$ molecules are indeed present in each of the two regions of density we observe for this protein.

## Immuno-labelling and cross-linking MS confirm the localization of VirD4 within the T4SS$_{3-10+D4}$ complex

We next sought to validate the location of VirD4 within the T4SS$_{3-10+D4}$ EM reconstruction using two complementary techniques: immuno-labelling by NS-EM and chemical cross-linking coupled to mass spectrometry (XL-MS).

Immuno-labelling by NS-EM provides a means by which a particular component in a large complex can be located within the electron density of that complex. In brief, an antibody targeting a particular component of a complex is reacted to the complex; the resulting antibody-bound complex is imaged by NS-EM. Class averages of antibody-bound complex particles are then compared to class averages of non-bound complexes. Extra electron density corresponding to the antibody should be observed where the antibody has bound and this electron density should be located within close proximity of the component to which the antibody has bound. To immuno-label TrwB$_{/VirD4}$ within the larger T4SS$_{3-10+D4}$ complex, we generated a variant of the $\Delta trwD_{/virB11}$ construct with a FLAG tag-encoding sequence introduced at residue Thr236 of TrwB$_{/VirD4}$. The 236 position in the protein structure is indicated by a red sphere in Fig 5A. This region belongs to the cytoplasmic domain of

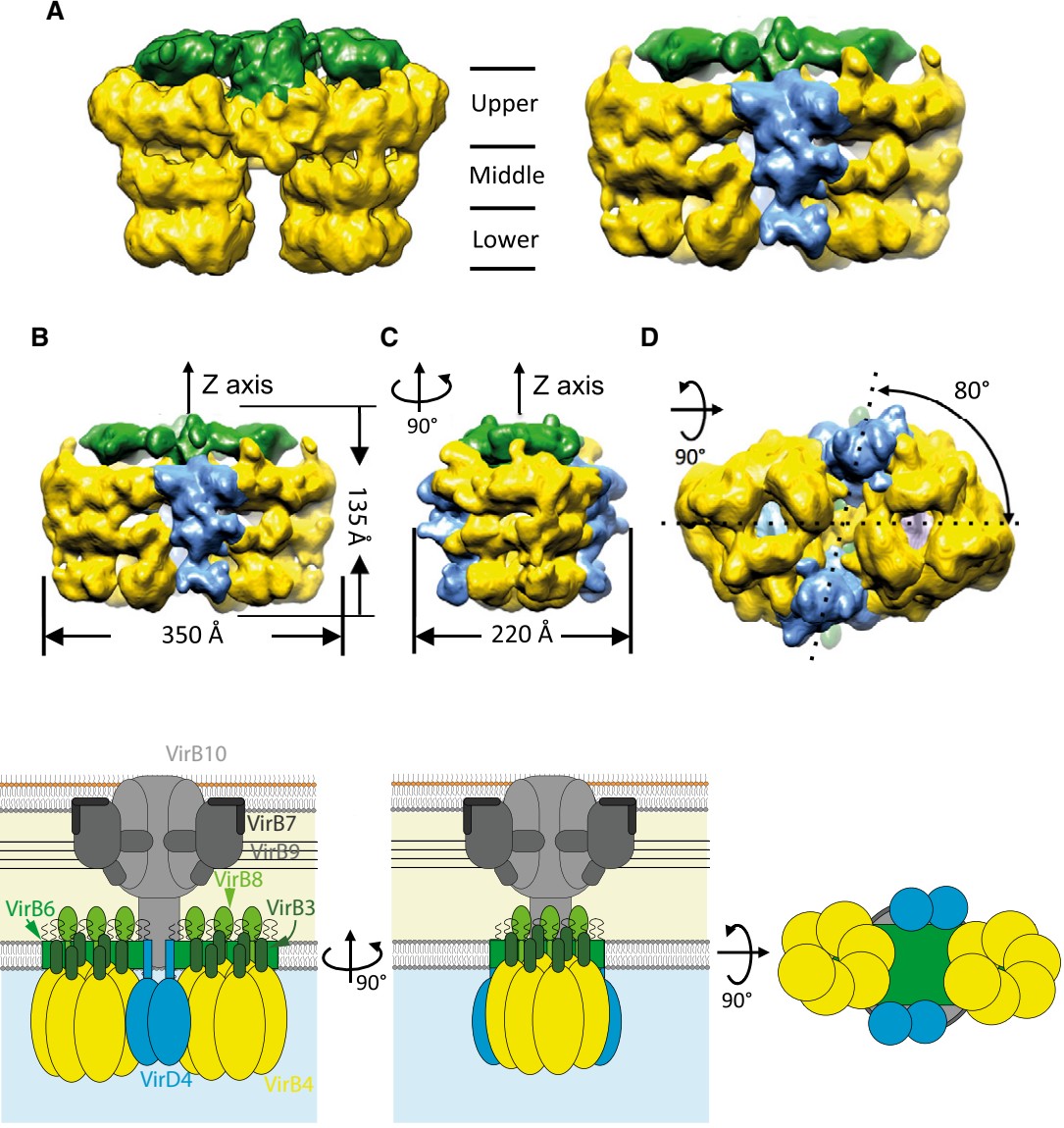

**Figure 4.  NS-EM structure of the T4SS$_{3-10+D4}$ complex.**

A   Side views of the NS-EM structure of the T4SS$_{3-10}$ (left) and T4SS$_{3-10+D4}$ (right) complexes. The arches, the TrwK$_{/VirB4}$ barrels and the TrwB$_{/VirD4}$ density are shown in green, yellow and blue, respectively. The boundaries of the three-tier structure of the TrwK$_{/VirB4}$ barrel are indicated in the middle of the panels and each tier is labelled as lower, middle and upper, respectively.

B   Side view of the T4SS$_{3-10+D4}$ structure (upper panel) and derived schematic representation (lower panel). Overall dimensions are indicated.

C   Same as in (B) except that the side view shown is rotated 90 degrees along a vertical axis compared to the view shown in panel (B). Overall dimension is indicated.

D   Bottom view of the NS-EM structure of the T4SS$_{3-10}$ (upper panel) and the corresponding schematic representation (lower panel). Angle between the axes connecting the TrwK$_{/VirB4}$ hexamer pair and the TrwB$_{/VirD4}$ dimer pair is shown.

TrwB$_{/VirD4}$ and is at the opposite end of the structure relative to the N-terminus which is known to be inserted in the inner membrane. Therefore, the FLAG tag inserted in this region should be easily accessible for binding by anti-FLAG antibodies and, if our localization of TrwB$_{/VirD4}$ as seen in the NS-EM structure of the T4SS$_{3-10+D4}$ complex (Fig 4) is correct, the bound antibodies should be observed on the side or at the bottom of the IMC in the NS-EM class averages of the antibody-bound complex.

The FLAG-tagged complex termed "T4SS$_{3-10+D4FLAG}$" was purified and reacted to anti-FLAG antibodies. The antibody-bound complex was further purified, applied to a grid and stained using negative stain as described in Materials and Methods. After data collection, class averages were obtained: these clearly show extra density for the FLAG antibody in close proximity of TrwB$_{/VirD4}$, either on the side or near the bottom (Fig 5B). Difference density (Fig EV3A, right panels) between class averages of the T4SS$_{3-10+D4}$ (Fig EV3A, left panels) and antibody-reacted T4SS$_{3-10+D4FLAG}$ (Fig EV3A, middle panels) clearly illustrates the presence of the extra density generated by antibody-binding. Thus, the additional density observed by comparing

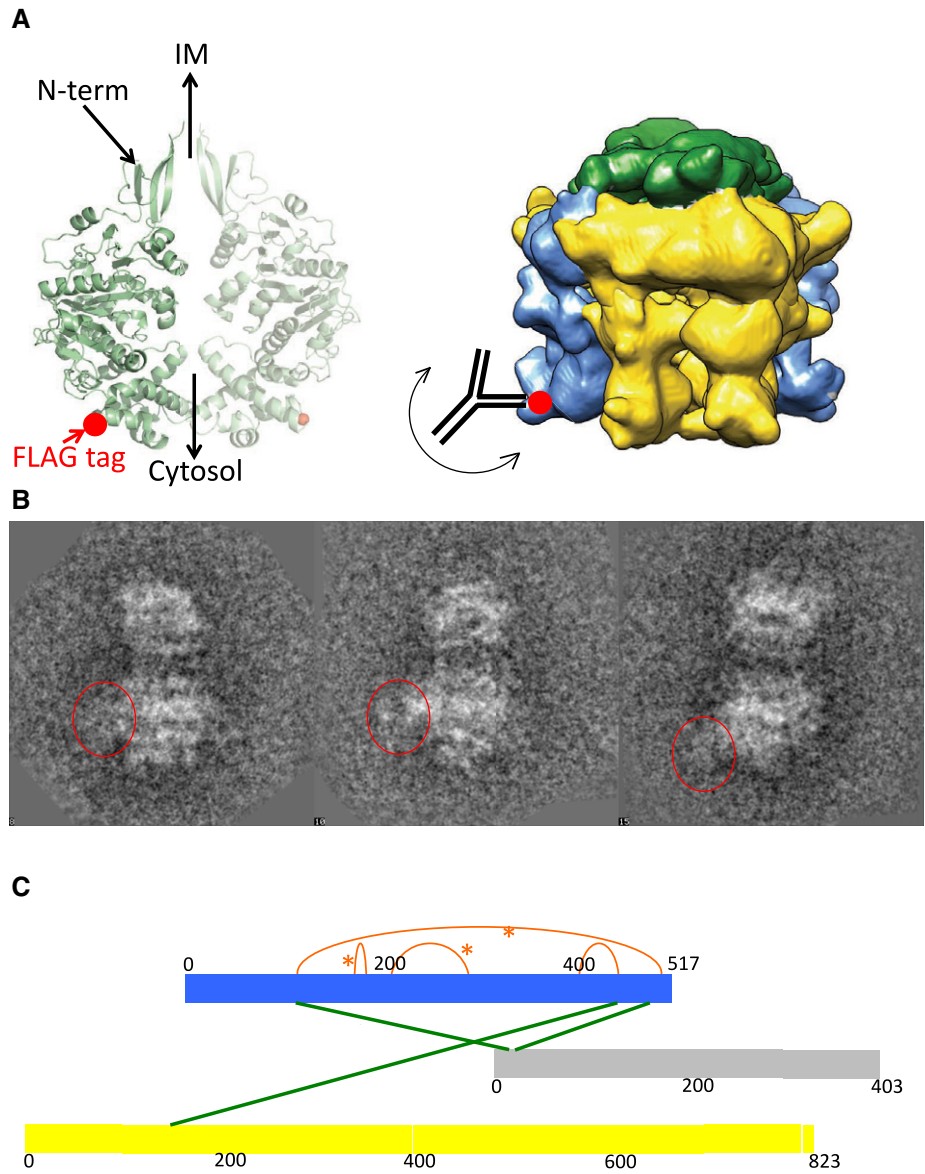

**Figure 5.  Validation of the location of TrwB$_{/VirD4}$ in the T4SS$_{3-10+D4}$ complex.**

A   Location of position 236 of TrwB$_{/VirD4}$ where a FLAG tag was introduced for immuno-labelling of TrwB$_{/VirD4}$ within the T4SS$_{3-10+D4}$ complex. Left: structure of hexameric TrwBΔN70$_{/VirD4ΔN70}$ (PDB entry code 1GKI). Only two diametrically opposed subunits are shown. The position of residue 236 is shown with a red sphere. Right: region of the T4SS$_{3-10+D4}$ IMC where the location of the residue 236 of TrwB$_{/VirD4}$ would be expected (indicated by a red sphere) and schematic representation of an anti-FLAG antibody bound to that location.

B   Class averages of antibody-bound T4SS$_{3-10+D4FLAG}$. Class averages of the aligned NS-EM images of the T4SS$_{3-10+D4FLAG}$ with 3–4 particles per class. Red circles indicate the position of the antibody.

C   Network of spatial restraints identified by XL-MS mapped onto primary sequences of the TrwB$_{/VirD4}$ subunit (blue) and the subunits interacting with it, TrwE$_{/VirB10}$ (grey) and TrwK$_{/VirB4}$ (yellow). Inter-molecular and intra-molecular cross-links are shown in green and orange, respectively. *Indicates cross-links mapped in the structure shown in Fig EV3B.

the T4SS$_{3-10}$ and T4SS$_{3-10+D4}$ (Fig 4) can be indeed attributed to TrwB$_{/VirD4}$.

For XL-MS, the purified T4SS$_{3-10+D4}$ complex was cross-linked using amine-to-amine cross-linkers (BS$^3$ and DSS), which connect lysine residues that are located within a maximum distance of 33–35 Å (Matthew Allen Bullock *et al*, 2016). We used a protocol similar to that described by Leitner *et al* (2014) to perform the XL-MS experiments and applied stringent criteria during the filtering of

the database search results in order to eliminate false positives. Using this approach, we identified a total of 41 unique cross-links (26 inter-protein and 15 intra-protein). Of particular interest for the study presented here are the seven cross-links involving TrwB$_{/VirD4}$ (four intra-molecular and three inter-molecular; Appendix Table S2). Out of the four intra-molecular cross-links, three can be mapped onto the crystal structures of R388 TrwB$_{/VirD4}$ (Fig EV3B). The fourth cross-link could not be mapped onto the

structure as it involves a Lys residue in a disordered loop (PDB: 1GKI; Fig EV3B). Of the three inter-molecular cross-links, two were between TrwB$_{/VirD4}$ and TrwE$_{/VirB10}$, a known interaction, and one was with TrwK$_{/VirB4}$ (Fig 5C). The observed cross-links between TrwB$_{/VirD4}$ and TrwE$_{/VirB10}$ or TrwB$_{/VirD4}$ and TrwK$_{/VirB4}$ are consistent with structural knowledge since the residues involved in the cross-links are all likely to locate near the cytosolic side of the inner membrane. Proximity of TrwB$_{/VirD4}$ to TrwK$_{/VirB4}$ was therefore confirmed, consistent with the results of the NS-EM structure.

## Conclusion

Coupling proteins in conjugative T4SSs play important roles in recruiting the substrate to the T4S machinery (Cabezon *et al*, 1997; Cascales & Christie, 2004; Lang *et al*, 2010, 2011). Although a lot of information has been gathered over the years about the activity, oligomerization state and structure as well as the importance of VirD4 with respect to substrate recognition and processing, its location within the T4S system was unknown (Cascales & Christie, 2004; Mihajlovic *et al*, 2009; Lang *et al*, 2010; de Paz *et al*, 2010; Whitaker *et al*, 2016). This is the issue we set out to address in this study. In order to achieve this, we designed functional expression systems that resulted in TrwB$_{/VirD4}$ forming a stable complex with the previously investigated T4SS$_{3-10}$ complex (Low *et al*, 2014). This interaction is primarily mediated by the core OMC complex as deletion of TrwM$_{/VirB3}$ or TrwK$_{/VirB4}$ results in the formation of a complex between TrwB$_{/VirD4}$ and the core OMC complex alone. It is known that TrwB$_{/VirD4}$ and one essential core OMC component, TrwE$_{/VirB10}$, interact, but it is nevertheless striking that this is likely the most important interaction recruiting TrwB$_{/VirD4}$ to the T4S system as all other core OMC components are either periplasmic or associated with the outer membrane.

The structure of the T4SS$_{3-10+D4}$ sheds unprecedented light on the location of VirD4 within the T4S machinery. Two regions of extra electron density were present in the T4SS$_{3-10+D4}$ complex, which were absent in the T4SS$_{3-10}$ complex. These densities exhibit the three-layered structure observed for TrwK$_{/VirB4}$ in the T4SS$_{3-10}$ complex (Low *et al*, 2014), which is to be expected as VirD4 and VirB4 share many structural similarities (Wallden *et al*, 2012). The two TrwB$_{/VirD4}$ patches of density are located opposite to one another, do not connect, and the TrwB$_{/VirD4}$ pair is at a ~80° angle from the pair of TrwK$_{/VirB4}$ barrels (Fig 4D). Hence, the duplication of the VirB4 ATPase observed in the T4SS$_{3-10}$ complex appears to be mirrored by the VirD4 ATPase. Thus, given that there are, in total, three different ATPases (VirB4, VirD4 and VirB11) mediating type IV secretion, the finally assembled T4S system may comprise six hexameric ATPases at any one time, an extraordinary situation, and, to our knowledge, unique in biology.

However, TrwB$_{/VirD4}$ is not represented within our structure as a hexamer: the electron density is only sufficient for two TrwB$_{/VirD4}$ monomers (Fig EV2C). Although as an AAA+ ATPase VirD4 is expected to form a hexamer, it has been shown previously that TrwB$_{/VirD4}$ can be purified in a monomeric form and only oligomerizes in the presence of DNA (Matilla *et al*, 2010). It has been also hypothesized that VirD4 may bind to the T4S system as a monomer and only oligomerizes upon substrate recruitment and binding

(Cabezon & de la Cruz, 2006). Therefore, the structure of the T4SS$_{3-10+D4}$ complex might represent the structure of the initiation T4SS complex in its resting state, waiting for the substrate to be delivered and activate the secretion machinery.

The close proximity of VirD4 and VirB4 might provide the most appropriate functional platform to couple protein and DNA transport. Indeed, it is known that transfer of DNA from a donor cell to a recipient cell through the T4S system involves a covalent protein–ssDNA complex where both, the protein and the ssDNA need to be transferred (Zechner *et al*, 2012; Ilangovan *et al*, 2015). The proteinaceous part of this protein–ssDNA substrate is the relaxase. The relaxase nicks the plasmid DNA to be transferred at *oriT* and covalently binds to the resulting 5′-phosphate: it is this relaxase–ssDNA complex that undergoes T4S system-mediated transport. VirB4 has previously been described as a protein translocase while VirD4 is thought to be involved in ssDNA transport (Atmakuri *et al*, 2004). It is therefore tempting to speculate that the relaxase part of the substrate is transported through VirB4 while the ssDNA part of the substrate is handled by VirD4: in that context, the proximity of the two ATPases as observed in the T4SS$_{3-10+D4}$ complex would make sense.

## Materials and Methods

### Molecular biology

The strains and primers used in this study are listed in Appendix Tables S3 and S4, respectively. Unless stated otherwise, plasmids used in this study (Appendix Table S1) have been generated by following the protocol described in the IN Fusion HD Cloning kit from Clontech.

To generate pBADM11_*trwN$_{/virB1}$*-*trwE$_{/virB10Strep}$*_*trwD$_{/virB11}$*, because the ribosome-binding site (RBS) sequence for *trwD$_{/virB11}$* is contained within the sequence encoding the C-terminus of TrwE$_{/VirB10}$, a consensus *Escherichia coli* RBS sequence was cloned between the stop codon for *trwE$_{/virB10}$* and the methionine-encoding ATG of *trwD$_{/virB11}$*. Using this construct, two additional clones were produced, one with the TrwB$_{/VirD4}$-encoding gene (*trwB$_{/virD4}$*) located after the sequence encoding TrwD$_{/VirB11}$ and the other with the TrwB$_{/VirD4}$-encoding gene inserted before the sequence encoding TrwN$_{/VirB1}$. In the former, the sequence encoding a 10-Histidine tag was added at the 5′ end of the *trwB$_{/virD4}$* sequence so as to produce an N-terminally His-tagged version of VirD4, while, in the latter, the same sequence was added at the 3′ end of the *trwB$_{/virD4}$* gene so as to produce a C-terminally His-tagged version of the protein. Also, in the former, expression of TrwB$_{/VirD4}$ relies on its own RBS (that of *trwB$_{/virD4}$*) while, in the latter, expression of TrwB$_{/VirD4}$ relies on the vector's RBS and expression of TrwN$_{/VirB1}$ relies on a consensus *E. coli* RBS sequence that was added. The clone with the TrwB$_{/VirD4}$-encoding sequence cloned after the TrwD$_{/VirB11}$-encoding sequence is named pBADM11_*trwN$_{/virB1}$*-*trwE$_{/virB10Strep}$*_*trwD$_{/virB11-His}$trwB$_{/virD4}$* while the clone with the TrwB$_{/VirD4}$-encoding sequence cloned before the TrwN$_{/VirB1}$-encoding sequence is named pBADM11_*trwB$_{/virD4His}$*_*trwN$_{/virB1}$*-*trwE$_{/virB10Strep}$*_*trwD$_{/virB11}$*.

To generate the pCDF_*trwABC* plasmid the *trwABC* operon was amplified, introducing XhoI and AscI restriction sites to the 5′ and 3′ ends, respectively. The PCR product was purified, digested with

XhoI and AscI, purified again and ligated into XhoI/AscI digested pCDFDuet-1, creating the plasmid pCDF_trwABC.

To generate the pCDF_trwAC plasmid, the *trwB* gene was knockout from the pCDF_trwABC plasmid by amplification of the entire vector except for the region encoding *trwB* from its 5′ end to the sequence encoding the RBS of *trwC* which is encoded within the 3′ end of the *trwB* gene.

To generate the pRSF*oriT* plasmid, the *oriT* region was amplified, introducing AscI restriction sites to both the 5′ and 3′ ends. The PCR product was purified, digested with AscI, purified again and ligated into AscI digested pRSFDuet-1, creating the plasmid pRSF*oriT*.

To generate the deletion mutants used in this study, single genes of $trwL_{/virB2}$, $trwM_{/virB3}$, $trwK_{/virB4}$, $trwJ_{/virB5}$, $trwI_{/virB6}$, $trwG_{/virB8}$ or $trwD_{/virB11}$ were deleted from the $pBADM11\_trwN_{/virB1}\text{-}trwE_{/virB10Strep\_}trwD_{/virB11\_His}trwB_{/virD4}$ plasmid. In the $\Delta trwL_{/virB2}$ construct, the first 35 nucleotides of the $trwL_{/virB2}$ gene were maintained, as this is the overlapping sequence of the *korA* gene, which is to the 5′ end of the $trwL_{/virB2}$ gene. In the $\Delta trwM_{/virB3}$ construct, the last 10 nucleotides of the $trwM_{/virB3}$ gene were maintained, as this sequence contains the potential RBS sequence of the $trwK_{/virB4}$ gene. In the $\Delta trwK_{/virB4}$ construct, the last 15 nucleotides of the $trwK_{/virB4}$ gene were maintained, as this sequence contains the potential RBS sequence and the start codon of the $trwJ/virB5$ gene. In the $\Delta trwJ_{/virB5}$ construct, the first 4 and the last 23 nucleotides of the $trwJ_{/virB5}$ gene were maintained, as this sequence contains the stop codon of the $trwK_{/virB4}$ gene and the potential RBS sequence and the start codon of the *eex* gene, respectively. In the $\Delta trwG_{/virB8}$ construct, the first 4 nucleotides of the $trwG_{/virB8}$ gene were maintained, as this sequence contains the stop codon of the $trwH_{/virB7}$ gene.

To generate the various FLAG-tagged variants used in this study, the $pBADM11\_trwN_{/virB1}\text{-}trwE_{/virB10Strep\_}trwD_{/virB11\_His}trwB_{/virD4}$ and $\Delta trwX_{/virBY}$ constructs were used as templates. In order to create a construct where $TrwM_{/VirB3}$ or $TrwG_{/VirB8}$ possessed a FLAG tag at the N-terminus, the nucleotide sequence encoding the FLAG tag was introduced at the 5′ end of the $trwM_{/virB3}$ and $trwG_{/virB8}$ genes. In the constructs with the FLAG tag incorporated at the C-terminus of the $TrwK_{/VirB4}$, $TrwJ_{/VirB5}$ and $TrwI_{/VirB6}$, the nucleotide sequence encoding for the FLAG tag has been inserted at the 3′ end of the $trwK_{/virB4}$, $trwJ_{/virB5}$ and $trwI_{/virB6}$ genes.

To generate the plasmid to produce $TrwB\Delta N70_{/VirD4\Delta N70}$ for antibody production, the $trwB\Delta N70_{/virD4\Delta N70}$ fragment was amplified and inserted into pBADM11 vector.

To generate the plasmid expressing the $T4SS_{3\text{-}10+D4FLAG}$ complex where a FLAG tag sequence is inserted in the loop of the $TrwB_{/VirD4}$ at Thr236, the $pBADM11\_trwN_{/virB1}\text{-}trwE_{/virB10Strep\_}trwD_{/virB11\_His}trwB_{/virD4}$ vector was linearized using primers, which contained the FLAG tag sequence.

## Conjugation assay

Overnight cultures of donor cells, in this study *E. coli* TOP10 cells with plasmids carrying T4S system component genes, relaxosome genes and *oriT* sequence as well as recipient strain, *E. coli* DH5α, were diluted and grown at 37°C with appropriate antibiotics to an $OD_{600}$ of 0.5. Expression of genes required for the DNA transfer was induced by addition of 0.08% w/v arabinose and 1 mM IPTG. Cells were incubated for another hour prior to performing the conjugation

assay. A total of 1.5 ml of donor culture, and the equivalent amount of recipient culture according to $OD_{600}$ measurements were washed clean of antibiotics using LB media and filtered on a hydrophilic MF-Millipore Membrane (Merck Millipore), made of mixed cellulose esters with a pore size of 0.45 μm. The filters were then incubated for 1.5 h on a LB agar plate with cells facing upwards. The cells were subsequently washed off the filter paper using 1 ml LB, and 100 μl of different dilutions was plated on LB agar plates containing nalidixic acid (Nal, selecting for *E. coli* DH5α recipient strains) and a second antibiotic, whose resistance is encoded on the transported plasmid (kanamycin (Kan) in this study). Appropriate dilutions of the conjugation culture were also plated on antibiotics selecting for donors or total *E. coli* DH5α recipients. The transfer efficiency was calculated by dividing transconjugant colony count by recipient colony count (Cellini *et al*, 1997).

## Purification of the T4SS$_{3\text{-}10+D4}$ complex, single-component deletion mutants and FLAG-tagged T4SS$_{3\text{-}10+D4FLAG}$ variant

The T4SS$_{3\text{-}10+D4}$ complex or the single-component deletion mutants of the T4SS$_{3\text{-}10+D4}$ complex were purified according to the following procedure. Overnight culture of a freshly transformed TOP10 cells (Thermo Fischer) were diluted into six litre of LB medium supplemented with 100 μg/ml of carbenicillin. Cells were grown at 37°C with shaking. When the $OD_{600}$ of the cell culture reached 0.5, cells were induced by addition of 0.08% arabinose and the expression was carried on at 16°C with shaking overnight. Cells were pelleted by spinning down for 30 min at $5,000 \times g$ and resuspended in lysis buffer: 50 mM HEPES pH = 7.6, 200 mM sodium acetate, 1 mM EDTA pH = 8.0 with a protease inhibitor cocktail tablet (Roche), 100 μg/ml lysozyme and 1 μg/ml DNaseI. After stirring of the mixture on ice for 15 min, the cells were lysed by passing them twice through an Emulsiflex C-5. Cell debris was separated from the supernatant by spinning for 30 min at $26,712 \times g$. Membranes were pelleted by ultracentrifugation of the supernatant for 45 min at $95,834 \times g$ and manually homogenized using solubilization buffer: 50 mM HEPES pH = 7.6, 200 mM sodium acetate, 1 mM EDTA pH = 8.0, 0.5% w/v n-dodecyl-β-D-maltopyranoside (DDM) (Anatrace), 0.5% digitonin (Sigma-Aldrich), 0.05 mM n-tetradecyl-N,N-dimethylamine-N-oxide (TDAO) (Anatrace), 2 mM tris(2-carboxy-ethyl)phosphine (TCEP) (Sigma-Aldrich) and placed on the rotary shaker at 4°C for 1 h. Insoluble materials were separated by another ultracentrifugation for 20 min at $95,834 \times g$. The supernatant was loaded to a 5 ml Strep column (GE Healthcare) equilibrated with buffer A: 50 mM HEPES pH = 7.6, 200 mM sodium acetate, 0.1% digitonin, 0.05 mM TDAO. After extensive washing with buffer A, samples were eluted with buffer A + 2 mM desthiobiotin and 50 mM imidazole pH = 7.6 directly into 2 × 1 ml HisTrap columns (GE Healthcare) equilibrated with the same buffer. After additional washing step with buffer A containing 100 mM imidazole pH = 7.6, the protein complex was eluted with buffer A containing 350 mM imidazole pH = 7.6.

Sample of the T4SS$_{3\text{-}10+D4}$ complex used for the NS-EM experiments was in addition to the protocol mentioned above stabilized by mild cross-link with 0.1% of glutaraldehyde (Sigma-Aldrich) while being applied to the 10–30% sucrose density gradient at $50,512 \times g$ for 15 h at 4°C using a SW40Ti rotor (Stark, 2010). Samples from the gradient were fractionated and analysed by

SDS–PAGE. Fractions with bands at high molecular weight on the gels were analysed by NS-EM. Grids of the samples which looked the most concentrated and least aggregated on the NS-EM were used for collection of the images which were used for EM data processing.

### Electron microscopy and data processing

Negative stain electron microscopy (NS-EM) was used to analyse the complex. 5 µl of T4SS$_{3-10+D4}$ complex at concentration of around 0.03 mg/ml was applied on glow discharged carbon coated copper grids (type B 400, Agar Scientific) and incubated at room temperature (RT) for 3 min. The excess of the sample was blotted away and the grid was stained with 5 µl of 2% uranyl acetate for 3 min. The excess stain was then blotted away and the grid was dried at room temperature. Grids were imaged on a Tecnai F20 FEG microscope operating at 200 kV at a magnification of 41,500 using a defocus range of −0.5 to −2 µm. Images were recorded using the direct electron detector DE20 (1.54 Å per pixel) using a total dose of 30 electrons/Å$^2$. Frames from 2 to 29 were aligned using EMAN2 (Tang *et al*, 2007) and IMOD (Kremer *et al*, 1996) software. The CTF was estimated using CTFFIND3 (Mindell & Grigorieff, 2003) and correction was done by phase flipping using BSOFT (Heymann, 2001) on the aligned frames. A total of 14,918 particles were manually selected using XMIPP v3.0 (Scheres *et al*, 2008) and extracted from aligned, averaged and CTF corrected frames with a box size of 400 pixels. All subsequent operations were done using IMAGIC-5 (van Heel *et al*, 1996).

Images were normalized, band-pass filtered and subjected to multireference alignment (MRA) using re-projections (side views only) of the 3D reconstruction EMD-2567 of the T4SS$_{3-10}$ complex, followed by multivariate statistical analysis (MSA) (van Heel *et al*, 2000). The best classes representing characteristic views were used as new references for the next iteration of the alignment procedure. Aligned particle images were shifted up so the centre of the IMC was located in the centre of the frame, then a circular mask was applied to remove parts of the image corresponding to the outer membrane core complex and the surrounding noise. Then several rounds of MRA and MSA were run using re-projections of the IMC part of the T4SS$_{3-10}$ reconstruction (i.e. EMD-2567 where the outer membrane core complex was masked out). The best classes were selected according to the minimal deviation between images that compose the classes. Euler angles were assigned to these classes by angular reconstitution (Van Heel, 1987) using an anchor set calculated using the IMC part of the T4SS$_{3-10}$ reconstruction. No symmetry was applied during the first steps of the reconstruction. The first asymmetrical reconstruction clearly demonstrated features corresponding to C2 symmetry with a self cross-correlation of the non-symmetrized map of about 75%. During subsequent steps of refinement, the C2 symmetry was applied. To improve the alignment and angular assignment further rounds of MRA-MSA-angular reconstitution were run iteratively using the best classes for the reconstructions. Eventually, 2000 classes were used with around 7-8 particles per class during the refinement procedure. In the final structure, ~700 classes with ~4 images per class were used. The resolution of the 3D reconstruction was estimated by Fourier Shell Correlation (FSC) (van Heel & Schatz, 2005) as 28 Å using 0.5 criteria. Visualization of the final 3D map of the IMC was made

in Chimera (Pettersen *et al*, 2004). Segmentation of the T4SS$_{3-10+D4}$ complex was performed using Chimera.

### Determination of the complex stoichiometry by cysteine labelling with Alexa Fluor dye

50 µl (~0.3 mg/ml) of sample of the T4SS$_{3-10+D4}$ complex after elution from the HisTrap column was buffer exchanged using a PD 10 (GE Healthcare) desalting column to a buffer containing 50 mM HEPES pH 7.2, 200 mM sodium acetate and 1 mM EDTA. The complex was then denatured by boiling in 1% SDS. Disulphide bonds were reduced by incubation with TCEP at 2 mM concentration for 20 min at room temperature. Alexa Fluor® 633 C$_5$ maleimide (Thermo Fischer Scientific) was added to 200 µM final concentration and samples were incubated at room temperature for an additional 1 h protected from light. Proteins were separated using SDS–PAGE, and the gel was visualized using Image Reader FLA-3000. Intensity of the bands was analysed using the Image Gauche software.

### Immuno-labelling of the T4SS$_{3-10+D4FLAG}$ complex and visualization by NS-EM

T4SS$_{3-10+D4FLAG}$ complex was purified using the same protocol as described for the T4SS$_{3-10+D4}$ complex. Freshly prepared T4SS$_{3-10+D4FLAG}$ sample was incubated with anti-FLAG antibody produced in goat (Abcam) in a 1:5 molar ratio for 1 h on ice. The excess of the antibody was removed by loading the mixture to a 1 ml StrepHP column (GE Healthcare) equilibrated with buffer A. After washing the column with buffer A the complex with the antibody was eluted from the column using buffer A + 2 mM desthiobiotin. The presence of the antibody within the eluted complex was confirmed by Commasie stained SDS–PAGE gel and WB, using anti-goat antibodies conjugated with alkaline phosphatase (Abcam). The bands were visualized by incubating the membrane with SIGMA-FAST™ BCIP®/NBT (Sigma-Aldrich).

NS-EM grids of the T4SS$_{3-10+D4FLAG}$ with the anti-FLAG antibody were prepared as described for the T4SS$_{3-10+D4}$ complex. Grids were imaged on a Tecnai F20 FEG microscope operating at 200 kV at a magnification of 35 750 using defocus range −1.5 to −2 µm. Images were recorded using the direct detector DE20. Movies were collected (1.79 Å per pixel) using a total dose of 30 electrons/Å$^2$. Frames from 2 to 20 were aligned using EMAN2 (Tang *et al*, 2007) and IMOD (Kremer *et al*, 1996) software. The CTF was estimated using CTFFIND3 (Mindell & Grigorieff, 2003) and correction was done by phase flipping using BSOFT (Heymann, 2001) on the aligned frames. A total of 80 particles, representing complexes with and without a clear extra density corresponding to antibody, were manually picked from aligned, averaged and CTF corrected frames with a box size of 320 pixels. All subsequent operations were done using IMAGIC-5 (van Heel *et al*, 1996).

Images were normalized and aligned to one of the images. This was followed by MSA (van Heel *et al*, 2000). Next the images were classified based on the eigenvectors which showed the differences corresponding to the presence and absence of the antibody within the particles. To identify the binding site of the antibody within the complex, six classes with 3–4 particles per class were selected. Class averages of the particles with the antibody bound were compared

with particles within the same orientation to localize the extra density corresponding to the antibody. The difference between the two has been created as well.

### Purification of TrwBΔN70/VirD4ΔN70 for raising antibodies

Overnight culture of a BL21* cells (Thermo Fisher) freshly transformed with pBADM11$_{His\_TEV}$*trwB/virD4ΔN70* (Appendix Table S1) was diluted into six litre of LB medium supplemented with appropriate antibiotic. Cells have been grown at 37°C with shaking. When the $OD_{600}$ of the cell culture reached 0.5, cells were induced by the addition of 0.08% arabinose and the expression was carried out at 16°C with shaking overnight. Cells were pelleted by spinning down for 30 min at 5,000 × *g* and resuspended in lysis buffer: 50 mM HEPES 7.6, 250 mM NaCl, 1 mM EDTA with protease inhibitor cocktail tablet (Roche), 100 μg/ml lysozyme and 1 μg/ml DNaseI. After stirring of the mixture on ice for 15 min, the cells were lysed by passing them twice through Emulsiflex C-5. Cell debris was separated from the supernatant by ultracentrifugation for 45 min at 235,418 × *g*. Supernatant was filtered through 0.45-μm filter and loaded to the 5 ml HisTrap column equilibrated with loading buffer (50 mM HEPES 7.6, 250 mM NaCl, 100 mM imidazole). After washing the column proteins were eluted with an increasing concentration of imidazole up to 500 mM imidazole. Fractions containing the protein were pooled and dialysed overnight to the loading buffer. During the dialysis, the protein sample was incubated with 6xHis-TEV protease in mass ratio 1:50 of the protease to the protein. Cleaved protein was separated from the protease and the non-cleaved protein by passing the sample through a 5 ml HisTrap column equilibrated with loading buffer. Flow through from the column was collected and concentrated by dialysis to PBS buffer adjusted with 50% glycerol. Proteins were aliquoted and flash frozen and stored at −80°C.

### Antibody purification

Antibodies were purified based on manufacturer's protocol. Briefly, 1 ml HiTrap NHS-activated HP column (GE Healthcare) was activated by applying a drop of ice cold 1 mM HCl followed by slowly washing it with additional 6 ml of 1 mM HCl. 1 ml of protein sample (0.7 mg of TrwBΔN70/VirD4ΔN70), which has been buffer exchanged into coupling buffer (0.2 M NaHC0$_3$, 0.5 M NaCl, pH = 8.3), was immediately loaded to the column and incubated at room temperature for 30 min. Excess of uncoupled protein was removed by washing the column with 1 ml of coupling buffer. Column was then deactivated by sequential washes with 6 ml of buffer A (0.5 M ethanoloamine pH = 8.3), 6 ml of buffer B (0.1 M sodium acetate pH = 4.0, 0.5 M NaCl) and another 6 ml wash with buffer A. Column with protein was incubated at this step at room temperature for 30 min. Afterwards, the column was washed with 6 ml of buffer B, 6 ml of buffer A, 6 ml of buffer B and finally with 5 ml of PBS buffer. 20 ml of rabbit blood serum containing antibodies was loaded to the column overnight at 0.1–0.2 ml/min flow rate. The column was afterwards washed with 40 ml of PBS followed by additional 20 ml wash with PBS supplemented with 0.5 M NaCl and another 20 ml wash with PBS buffer. Antibodies were eluted from the column using elution buffer (100 mM glycine pH = 2.3). In order to immediately neutralize the very low pH of the buffer that

the samples are eluted in, 2 M Tris pH = 8.0 was added to the collection tubes before starting the fractionation. Fractions were analysed on the SDS–PAGE gel and the samples containing antibodies were pooled and dialysed against PBS buffer supplemented with 50% glycerol.

### Detection of the expression levels of TrwB/VirD4 within the different deletion mutants expressing R388 T4S system

Overnight culture of TOP10 strain freshly transformed with a plasmid carrying genes expressing either T4SS$_{3-10+D4}$ or T4SS$_{3-10+D4}$ lacking one of the T4S complex encoding genes were diluted into a LB medium supplemented with 100 μg/μl of carbenicillin and grown and 37°C at 185 rpm. When the $OD_{600}$ of the cultures reached 0.5, the expression of the genes was induced by addition of Arabinose at 0.08% w/v concentration. The expression was carried out at 18°C overnight. The equivalent of 1 ml of $OD_{600}$ 1 was collected from each sample and the cells were spun down for 5 min at 16,100 × *g*. Next, the cells were resuspended in NuPAGE® LDS Sample Buffer (Thermo Fischer) and lysed by passing them 15 times through a syringe equipped with a 0.5-μm needle. Sample was boiled at 95°C for 5 min and cooled down on ice before loading to the SDS–PAGE gel. In order to ensure that the signal quantified from the Western blot is in the linear range, samples of purified TrwBΔN70/VirD4ΔN70 at known concentration were loaded on the same gel as the samples of the overexpressed variants of the R388 T4S system. The antibodies used to detect the amount of expressed TrwB/VirD4 from different constructs were TrwBΔN70 antibody raised in rabbit and isolated in house in combination with rabbit anti-goat IgG (Thermo Fisher). The bands were visualized by incubating the membrane with SIGMAFAST™ DAB with metal enhancer (Sigma-Aldrich).

### Detection of the expression levels of FLAG-tagged T4SS components within the different deletion mutants expressing R388 T4S system

The samples have been prepared in the same way as described above for the detection of the TrwB/VirD4 expression. The only difference being that the expressed proteins were visualized by incubating the membrane with anti-FLAG antibody produced in rabbit (Abcam) followed by incubation with anti-rabbit antibody conjugated with horseradish peroxidase (Abcam). The bands were visualized by incubating the membrane with SIGMAFAST™ DAB with metal enhancer (Sigma-Aldrich).

### Cross-linking-mass spectrometry

#### *Chemical cross-linking of the purified T4SS$_{3-10+D4}$ complex*
50–150 μg of purified T4SS$_{3-10+D4}$ (at 0.1–0.5 mg/ml) was cross-linked either with 2.5–5 mM BS$^3$ d$_0$/d$_{12}$ mixture (Creative Molecules; dissolved freshly in 20 mM HEPES pH 7.6 at 50 mM) or 1–1.5 mM DSS d$_0$/d$_{12}$ mixture (Creative Molecules; dissolved freshly in DMF at 25 mM) at RT for 40 min with mild shaking. The reaction was quenched by addition of ammonium bicarbonate solution to a final concentration of 50 mM and the solvent was evaporated using a vacuum centrifuge. The cross-linking was confirmed by SDS–PAGE and the cross-linked sample was examined

by negative staining electron microscopy to ensure that the sample had not aggregated during the cross-linking reaction. The final set of cross-links was collected from six independent purifications.

### Sample preparation for LC-MS/MS

The cross-linked sample was prepared for LC-MS/MS using the following procedure (modified from Leitner *et al*, 2014): The sample was denatured by dissolving it in 8 M urea and 0.1% Rapigest (Waters) at 1 mg/ml. Disulphide bonds were reduced by 10 mM DTT at 37°C for 30 min and free cysteines were alkylated by 20 mM iodoacetamide at RT in the dark for another 30 min. The sample was then incubated with Lys-C (Wako Chemicals GmbH) at 1:50 (w/w) enzyme-to-substrate ratio for 2 h at 37°C with mild shaking. The sample was then diluted with 50 mM ammonium bicarbonate solution to a final concentration of 1 M urea and trypsin (Promega) was added at an enzyme-to-substrate ratio of 1:50 (w/w). The mixture was incubated overnight at 37°C with mild shaking. Following overnight digestion, the sample was acidified by adding formic acid to a final concentration of 2% (v/v). The sample was purified by solid-phase extraction (SPE) using 50 mg Sep-Pak tC18 cartridges (Waters). Briefly, the cartridge was activated with 500 µl of acetonitrile (ACN) and equilibrated with 1 ml of SPE wash solution (water/ACN/formic acid: 95/5/0.1 (v/v/v)). After loading the sample, the cartridge was washed twice with 500 µl of SPE wash solution and the sample was eluted with 500 µl of SPE elution solution (water/ACN/formic acid: 50/50/0.1 (v/v/v)). The eluted sample was concentrated by removing the solvent using a vacuum centrifuge. Following SPE, the sample was dissolved in 20 µl of SEC buffer (water/ACN/TFA: 70/30/0.1 (v/v/v)) and the cross-linked peptides were enriched using size-exclusion chromatography (SEC) by separation of peptides on a Superdex Peptide PC 3.2/300 column (pre-equilibrated with SEC buffer) using an AKTA Micro system. 100 µl collected fractions were concentrated by removing the solvent using a vacuum centrifuge. The SEC fractions were dissolved in MS buffer (water/ACN/TFA: 97/3/0.1 (v/v/v)) and analysed by LC-MS/MS.

### LC-MS/MS analysis

The collected fractions were analysed by LC-MS/MS using a nanoAcquity UPLC system (Waters) connected to LTQ-Orbitrap Velos instrument (ThermoFisher Scientific). Peptides were separated on an NTCC-360/100-5-153 column (Nikkyo Technos Co.) at a flow rate of 400 nl/min using the following gradient: 0–60 min, 3–40%B; 60–61 min, 40–85%B; 61–71 min, 85%B, where solvent A = 0.1% (v/v) formic acid in water and B = 0.1% (v/v) formic acid in ACN. The MS spectra were acquired in the Orbitrap (m/z range 400–2,000; 60,000 resolution), and up to 10 most abundant ions per scan were selected for fragmentation by collision-induced dissociation (normalized collision energy = 35, activation Q = 0.250) and analysed in the linear ion trap. Precursors of unknown charge state or charge states +1 and +2 were not selected for fragmentation, and selected precursors were added on a dynamic exclusion list for 10 s after one sequencing event. Samples were analysed in duplicates.

### XL-MS data analysis

Raw files were converted to the mzXML format using msconvert (Chambers *et al*, 2012) and then searched using xQuest (Leitner *et al*, 2014) against a database containing the sequences of VirB1 to VirB11 and VirD4. False discovery rates (FDR) were estimated using xProphet, and the results were filtered using the following parameters: FDR = 0.05, min delta score = 0.95, MS1 tolerance window = ±10 ppm, Id-score > 25, minimum number of cleavages per peptide = 3.

### Accession codes

The EM map of the $T4SS_{3-10+D4}$ complex has been deposited in the EMD data bank (http://www.ebi.ac.uk/pdbe/emdb) under EMDB entry code EMD-3585.

**Expanded View** for this article is available online.

### Acknowledgements

This work was supported by Wellcome Trust 098302 grant to GW. We would like to thank Prof Jasminka Godovac-Zimmermann and Dr Nigel Rendell for providing us access to the Orbitrap mass spectrometer used for the cross-linking experiments.

### Author contributions

AR, SC and GW designed the experiments; AR, MT and SC cloned all the constructs used in this study and AR and SC checked their functionality for DNA transfer; AR and MU purified the $T4SS_{3-10+D4}$ complexes and variants and determined stoichiometry; MU collected the NS-EM data set of the $T4SS_{3-10+D4}$ complex and together with EVO solved the structure of the $T4SS_{3-10+D4}$ complex; EVO supervised the EM analysis; AR performed the immuno-labelling experiments; AR and MT purified TrwB used for raising of the antibodies and purified the antibodies from the obtained serum; AR and CF-R purified the deletion constructs; MT performed XL-MS experiments and analysed the data with the help from AC and KT; AR, SC, RDH and GW wrote the manuscript.

### Conflict of interest

The authors declare that they have no conflict of interest.

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
