## [Review Process File · The EMBO Journal]

Manuscript EMBO-2017-96629

Structure of a VirD4 coupling protein bound to a VirB type IV secretion machinery

Adam Redzej, Marta Ukleja, Sarah Connery, Martina Trokter, Catarina Felisberto-Rodrigues, Adam Cryar, Konstantinos Thalassinos, Richard D. Hayward, Elena V. Orlova & Gabriel Waksman

Corresponding authors: Elena V. Orlova & Gabriel Waksman, Birkbeck College

Review timeline:

Submission date:	30 January 2017
Editorial Decision:	09 March 2017
Revision received:	05 July 2017
Editorial Decision:	03 August 2017
Revision received:	09 August 2017
Accepted:	22 August 2017

Editor: Ieva Gailite

Transaction Report:

1st Editorial Decision

09 March 2017

Thank you for submitting your manuscript for consideration by the EMBO Journal. We have now finally received all three reports on your manuscript, which I am copying below for your information.

As you can see from the comments, all three referees express interest in the presented structure of the type IV secretion system. However, they also raise substantive concerns with the analysis that would need to be addressed before they can support publication here. Based on the overall interest expressed in the reports I would like to invite you to submit a revised version of your manuscript in which you address the comments of all three referees. I would ask you to focus in particular on the following points:

- All referees indicate that the data on TrwB/VirD4 helical distribution are not conclusive. Please either expand this section substantively in line with referees' recommendations, or remove this section entirely.
- Provide further support to the location of TrwB/VirD4 in the structure (referee #2, point 4; referee #3, point 9).
- Please provide further support for the stoichiometry of the presented structure, as requested by referee #1 (point 2), referee #2 (point 1, 3), referee #3 (points 6,7).
- Provide complementation studies as requested by referee #1 (points 2A and 10)

I should add that it is The EMBO Journal policy to allow only a single major round of revision and that it is therefore important to resolve the main concerns raised at this stage. Since extensive additional work would be needed to fulfill all the referee requests, I would understand if you were to

choose not to undergo an extensive revision here and rather pursue a submission at an alternative venue, in which case please inform us about your decision at your earliest convenience.

When preparing your letter of response to the referees' comments, please bear in mind that this will form part of the Review Process File, and will therefore be available online to the community. For more details on our Transparent Editorial Process, please visit our website: http://emboj.embopress.org/about#Transparent_Process

We generally allow three months as standard revision time. Please contact us in advance if you would need an additional extension. As a matter of policy, competing manuscripts published during this period will not negatively impact on our assessment of the conceptual advance presented by your study. However, we request that you contact the editor as soon as possible upon publication of any related work, to discuss how to proceed.

Please feel free to contact me if have any further questions regarding the revision. Thank you for the opportunity to consider your work for publication. I look forward to your revision.

REFEREE REPORTS

Referee #1:

This study reports a number of findings of potential interest to the field of type IV secretion. The authors have attempted to integrate different lines of study into a cohesive report, but perhaps because of the breadth, certain sections are insufficiently developed. As discussed further below, the subcellular localization studies reporting a 'helical' pattern for the distribution of the Trw machine were particularly problematic. Another weakness was the lack of complementation of individual gene deletions; without these experiments no firm conclusions can be drawn regarding the contributions of a given Trw protein to machine assembly. Finally, the use of the 3-plasmid system is interesting and useful for these studies, but not intrinsically novel or, in fact, convincing since no data are presented showing that a single promoter drives the inducible expression of all of the Trw proteins. Results of the structural and mass spectrometry studies are the most compelling and interesting part of the manuscript, although it is curious that the isolated complexes yielded a stoichiometry of 5-8 VirD4(TrwB) monomers per machine while the same machine (isolated even with crosslinking) had only an associated VirD4(TrwB) dimer when analyzed by negative stain EM. The authors need to address the following comments.

Major:

1. Deficiencies need to be rectified with the 3-plasmid system.
 - A) Stated overproduction. The authors present no evidence that the Trw were overproduced when expressed from inducible promoters. Since only epitope-tagged proteins were evaluated, not the native proteins, no comparisons could be made among steady-state levels of epitope-tagged proteins produced from the inducible promoters and native proteins produced from the native promoter. The Llosa lab has antibodies to many of the Trw proteins, why not request those antibodies for direct comparisons?
 - B) Stated novelty: The authors overstate the novelty of their 3-plasmid system. In fact, such systems exist in nature and have also been previously engineered by other laboratories. For example, in nature genes encoding the T4SSs are generally expressed separately from coupling proteins - either on the same plasmid or on separate plasmids. See papers by Llosa et al. documenting transmissibility of plasmids carrying the R388 mob genes from the T4SS expressed from a separate plasmid. The capacity of T4SSs to mobilize oriT-containing plasmids or chromosomal fragments is also well established - see work from Meyer's lab as an example (there are many others). The authors should tamp down their claim of novelty and instead cite this large literature demonstrating the modular nature of T4SSs.
 - C) Why is the cloning of the trw locus behind inducible promoters, and then the subsequent demonstration of functionality considered 'reconstitution'? Basically, a locus was placed under the control of a regulatable promoter, this isn't reconstitution, it's simply cloning.

2. Effects of individual *trw* gene deletion on machine assembly.

A) The above points are also important because many *tra* loci are not expressed from a single upstream promoter and to this reviewer's knowledge there is no demonstration that all of the *trw* genes are expressed from a single promoter. Also, there are many instances in which deletion of a gene disrupts expression of upstream or downstream genes (e.g., due to translational coupling), but not other genes in the locus. Since the authors do not have antibodies to carefully address this question, minimally, complementation studies (by expression of the deleted gene from another plasmid) must be carried out to show that the deletion is directly responsible for altered protein content of the isolated subassemblies. Clearly, at this time, it is insufficient to show levels of VirD4(TrwB) expressed from the most distal gene in this locus, as its expression could be under the control of a constitutive internal promoter.

B) Some of the results presented in Figure 2 were already published by the Llosa lab (Llarea et al. 2013, Plasmid). Surprisingly this paper was not even cited - the paper should be cited and the present and previous findings compared.

C) Larrea et al. also reported that several *trw* mutations affect levels of TrwD(B11); the authors should attempt to reconcile why this protein is not copurified with their complexes.

3. The localization patterns observed for *gfp*- and FLAG-tagged VirB8(*trwG*) do seem to be similar to those seen by Zambryski for the Agrobacterium proteins. However, this is the weakest section of the manuscript for the following:

A) Data obtained with GFP-tagged protein cannot be relied on because the GFP tag disrupts protein function. Data obtained with the FLAG-tagged protein seem ok.

B) The authors do not show the localization pattern of GFP-B8 in the absence of the other machine subunits (although it is stated to be uniform - needs to be shown).

C) The colocalization of this protein with the T4SS would be more convincing if the spatial localization of more than one *Trw* protein were analyzed.

D) The comparison with Agrobacterium proteins is interesting, but in those studies the authors also almost exclusively analyzed attenuated or nonfunctional *gfp*-tagged proteins that were expressed from a plasmid other than that encoding the rest of the *Vir* proteins. It isn't clear why the authors think that the Agrobacterium system represents an 'endogenous' system while theirs is a 'reconstituted' system. Also, in fact the Agrobacterium system is a conjugation system, so it is wrong to state that the present study is the first to show a supposedly helical pattern for a conjugation system. It is interesting that both systems seem to show similar patterns, but that is the extent of the conclusions allowed by the available data.

E) The use of higher resolution microscopy (TIRF, SIM, cryoEM) has raised questions of whether even well-characterized proteins such as MreB form helical arrays or instead assemble as discrete disconnected patches. In view of this ongoing debate and based on the data presented, it is premature to conclude that this protein or the T4SS forms helical arrays.

Minor:

4. Pg. 5. Top. VirB4 and VirB11 are also required for substrate translocation - not just pilus biogenesis, as stated.

5. Pg. 5. Top. Considerable biochemical work was done on the VirB4 ATPases by Eric Lanka and others. These studies provided the first evidence for different oligomeric forms of these ATPases, and should be cited.

6. Pg. 7. Bottom. *trwA* and *ihf*. If these are references to bacterial proteins, they should have the first letter capitalized and not be italicized.

7. Pg. 11. Middle. The last sentence of this section makes no sense. The 'endogenous' system that needs to be compared is the R388 T4SS produced by R388 vs the inducible expression systems (which are not 'reconstituted' systems, but simply *tra* operons cloned behind inducible promoters). Overproduction of the *tra* proteins from the inducible promoters might lead to aberrant localization.

8. Pg. 11. What does "successfully integrate in the cell envelope" mean? This system promotes conjugative DNA transfer, so it must be successfully integrating into the cell envelope. The fluorescence microscopy data do not allow for a discrimination between 'successful' and 'nonsuccessful' integration.

9. Pg. 12. For the stoichiometric analyses, the gels should be shown in the supplementary files.

10. Pg. 14. Middle. Complementation studies need to be carried out to confirm that the deletions did not have polar effects on expression of genes either upstream or downstream of the mutation. This is

particularly important for mutations that appear to have disrupted machine production, e.g., virB3, virB4, virB8 deletions. Expression virB8 from another plasmid should both restore functionality of the T4SS and levels of VirB4. Similarly, expression of virB6 should restore VirB5 levels in the isolated machine. These studies are necessary to eliminate possible artifacts resulting from the disruption of adjacent gene expression through various mechanisms.

11. In view of the demonstrated interaction between TrwB and TrwE and its proposed importance, why not show that a trwE deletion mutation abolishes recovery of TrwB from the isolated machines?

12. Shifting between the Trw and Vir nomenclature confusing. This work focuses on the R388 system, so all references should be to the Trw proteins. A table could be included in the Suppl identifying the VirB homologs.

Referee #2:

This manuscript extends the current knowledge of the structure and assembly of the type IV secretion system, specifically addressing the location and interaction of the VirD4 protein with other components. VirD4 plays an important role in the recruitment of secreted substrate, and is located within the complex of VirB proteins through a combination of biophysical and structural analysis. As the title suggests, the results have implications for how secretion is energized by an assembly within the bacterial inner membrane which, intriguingly, comprises three different ATPases. The work has been carried out to a high technical standard.

Major

1. P6 third line from the bottom- do the authors actually '...determine the stoichiometry of all components within this complex' in this manuscript?
2. P10 and Fig 1 It would be useful to include the data on the distribution of fluorescence from eGFP-VirB8 alone, for comparison, to clarify the point made about localization of the reconstituted conjugative T4S systems into discrete foci.
3. P15 Line 9 What is the evidence that the barrel-like densities have 3-fold symmetry?
4. P15 The evidence for the location of VirD4 in the 3D volume is derived from differences in projections between T4SS3-10+D4 and T4SS3-10 in Fig 2. Was the 3D difference map also calculated? It would be clearer and more convincing to present that, in Fig 4A, for example. It should also be acknowledged somewhere that 3D volumes derived from negative stain are subject to flattening and distortion.
5. P18 Line 9 Can the authors be confident that the electron density only accommodates two VirD4 monomers and not, for example, one or three? Given previous comments on the limitations of negative stain-derived volumes and stoichiometry estimates, a suitable caveat should be applied to this statement.
6. Presumably the T4SS3-10+D4 3D volume will be deposited in the EMDB?

Minor

1. P6-7 The end of the Introduction contains an unnecessarily long summary of the main conclusions of the work; this could be pruned.
2. P7 7th line from top '...and located at..'
3. P9 8th line from top '...a 5-fold decrease...' According to Fig 1A, the mating efficiency of pAsk_virB1-11 is 0.4 and pAsk_virB1_B8eGFP-virB9-11 is 0.02, which is a 20-fold difference?
4. P10 Line 15 should 'pAsk_virB1-7_eGFPvirB8-11' be 'pAsk_virB1_B8eGFP-virB9-11'?
5. Figure 2: mass markers are needed for the gel in Fig 2B.

Referee #3:

In this manuscript Redzej and coauthors describe the negative stain electron microscopy structure of a complex comprising the TrwE-K, TrwM and TrwB proteins of the R388 conjugative Type IV secretion system. Compared to the previous structure of the TrwE-K-TrwM complex published by the same group in 2014, the TrwB coupling protein has now been added to the picture.

Comments.

- I do not understand the rationale to use the VirB nomenclature. The VirB proteins are the homologs in the Agrobacterium system, and it remains possible that the VirB system is structurally different. The Trw nomenclature (used in all the publications on this system since 1990!) should be used.

- Mating experiments. A negative control should be provided (pAsk and pBADM11 plasmids expressing TrwD-N with the TrwAC relaxosome plasmid).

- Some of the data presented are of poor quality, specifically the fluorescence microscopy experiments showed in Fig. 1 which have a very low resolution. Note that the presentation of these fluorescence experiments is sloppy as no statistical analysis is provided nor scale bars.

- The authors claim that the machinery locates on a helical pattern but the deconvolution images are not convincing - additional statistical analyses should be performed. The conclusion is also overstated as the authors imaged GFP-TrwG and there is no evidence that the discrete foci corresponds to assembled machinery rather than TrwG hetero-oligomeric complexes. In addition the R388 conjugative system is overproduced and the localisation observed may not reflect the real situation.

- Fig. 2B. TrwB associates with the different complexes lacking one subunit. However, TrwE is not tested as the purification tag is fused to this subunit. The authors should test a construct in which the TrwE transmembrane helix has been deleted.

- Fig. 2B. The authors claim that TrwJ is no longer present in the complex in the absence of TrwI. However, a SDS-PAGE or western blot analysis of the cell lysate should be provided. Is TrwJ produced?

- The data regarding the Alexafluor633 labeling to report the stoichiometry should be shown. The standard deviation obtained ($\pm 60\%$) is rather huge. The stoichiometry of TrwB in the complex should be addressed by an independent approach such as native mass spectrometry.

- Fig. 3C. It is not clear whether the TrwE-K/TrwM complex has been subjected to GraFix too.

- Whereas the location of TrwB is defined based on comparison with TrwBless complexes, a definitive demonstration is lacking such as immunogold labeling or purification of TrwE-K/M complex with TrwB-GFP.

1st Revision - authors' response

05 July 2017

Please find enclosed a revised version of our manuscript entitled "**Energizing the T4S system: structure of a VirD4 coupling protein bound to a VirB machinery**" by Adam Redzej, Sarah Connery, Marta Ukleja, Martina Trokter, Catarina Felisberto-Rodrigues, Adam Cryar, Konstantinos Thalassinos, Richard D. Hayward, Elena V. Orlova and Gabriel Waksman. The reviewers have provided a thorough review of the paper for which we are very grateful. Overall, the reviewers believe an important issue in the field of secretion is being addressed. They have however major concerns which we believe are entirely justified. In an excellent summary of what needs to be done to improve the manuscript in view of the reviewers' comments, you have asked us to focus on the following points:

1- All referees indicate that the data on TrwB/VirD4 helical distribution are not conclusive. Please either expand this section substantively in line with referees' recommendations, or remove this section entirely.

2- Provide further support to the location of TrwB/VirD4 in the structure (referee #2, point 4; referee #3, point 9).

3- Please provide further support for the stoichiometry of the presented structure, as requested by referee #1 (point 2), referee #2 (point 1, 3), referee #3 (points 6,7).

4- Provide complementation studies as requested by referee #1 (points 2A and 10).

You will find here an extensive point-by-point response to all reviewers' comments. But, first, we would like to describe how we have responded to your suggestions.

1- All referees indicate that the data on TrwB/VirD4 helical distribution are not conclusive. Please either expand this section substantively in line with referees' recommendations, or remove this section entirely.

Response. This section was considered weak by all reviewers. Our original intention was to draw attention to the fact that the multi-plasmid system we have used produced functional type IV secretion systems. However, as Reviewer 1 points out, other publications have already validated multi-plasmid systems for conjugation. Therefore, we have decided to focus on what Reviewer 1 says "is the most compelling and interesting part of the manuscript, i.e. the structural and mass spectrometry part of the paper. We have therefore taken up the Editor's suggestion of removing this section entirely. We would also like to take the opportunity at this stage to sincerely apologize for appearing to have ignored the Larrea et al. paper. We were very much aware of the data reported in this article and we have now cited it and compared the present and previous findings as suggested by Reviewer 1.

2- Provide further support to the location of TrwB/VirD4 in the structure (referee #2, point 4; referee #3, point 9).

Response.

We now provide further support for the location of TrwB/VirD4. Using immuno-labelling followed by visualisation by electron microscopy, we were indeed able to confirm independently the location of TrwB/VirD4. Immuno-labelling by NS-EM provides a means by which a particular component in a large complex can be located within the electron density of that complex. In brief, an antibody targeting a particular component of a complex is reacted to the complex; the resulting antibody-bound complex is then imaged by NS-EM and class averages of antibody-bound complex particles is compared to class averages of non-reacted complexes. Extra electron density corresponding to the antibody should be observed where the antibody has bound and this electron density should be located within close proximity of the component to which the antibody has bound.

To immuno-label TrwB/VirD4 within the larger T4SS_{3-10+D4} complex, we generated a variant of the *AtrwD/virB11* construct with a FLAG tag-encoding sequence introduced at residue Thr236 of TrwB/VirD4. The 236 position in the protein structure is shown as a red sphere in the new Figure 5A of this revised version of the manuscript. This region belongs to the cytoplasmic domain of TrwB/VirD4 and is at the opposite end of the structure relative to the N-terminus which is known to be inserted in the inner membrane. Therefore, a FLAG tag inserted in this region should be easily accessible for binding by anti-FLAG antibodies and, if our localisation of TrwB/VirD4 as seen in the NS-EM structure of the T4SS_{3-10+D4} complex (Figure 4) is correct, the bound antibodies should be observed on the side or at the bottom of the IMC in the NS-EM class averages of the antibody-bound complex.

The FLAG-tagged complex, termed "T4SS_{3-10+D4FLAG}" was purified and reacted to anti-FLAG antibodies. The antibody-bound complex was further purified, applied to a grid and stained using negative stain as described in Materials and Methods. After data collection, class averages were obtained: these clearly show extra-density for the FLAG-antibody in close proximity of TrwB/VirD4, either on the side or near the bottom (new Figure 5B). Difference density (new Figure S3A, right panels) between class averages of the T4SS_{3-10+D4} (new Figure S3A, left panels) and antibody-bound T4SS_{3-10+D4FLAG} (Figure S3A, middle panel) clearly illustrates the presence of the extra density generated by antibody-binding. Thus, the additional density observed by comparing the T4SS₃₋₁₀ and T4SS_{3-10+D4} (new Figure 4) can be indeed safely attributed to TrwB/VirD4.

3- Please provide further support for the stoichiometry of the presented structure, as requested by referee #1 (point 2), referee #2 (point 1, 3), referee #3 (points 6,7).

Response

Only reviewer 3 requests independent confirmation of the stoichiometry measurements using a different method than Cys-labelling. He/she suggests we should use native mass spectrometry.

However, this is a suggestion technically impossible to implement because we are dealing here with a complex which is over 3 MegaDalton in size and solubilised in detergent: no mass spectrometry method yet exists that would allow the desorption of such a large detergent-solubilised complex within the mass spectrometer. However, we have revisited our stoichiometry measurement and we now present better results with much smaller errors. We indeed redid the Cys-labelling experiment in triplicate and obtained a result much closer to the one derived from fitting TrwB/VirD4 in the electron density. While each of the two electron density patches for TrwB/VirD4 can only accommodate two copies of the protein, our previous stoichiometry measurements indicated a copy number of 5.3 with a large error of ± 3.0 . After having redone these experiments, we now have measured a copy number of 4.5 (within the range of the previous measurements) but with a much smaller variance of ± 0.2 . Since this latest measurement is more accurate, we now report it. It is also closer to the stoichiometry of 4 suggested by the size of the electron density attributed to TrwB/VirD4.

4- Provide complementation studies as requested by referee #1 (points 2A and 10).

Response

Complementation studies have already been reported by Larrea et al. (2013) on the same system where it is shown that transposon insertions in each of the *trw/virB* genes do not show polar effects except for the disruption of *trwL/virB2* that was shown to have a polar effect on expression of TrwK/VirB4. In the previous version of the manuscript, we had unfortunately omitted to cite this paper: we sincerely apologise for this oversight: it is a wonderful investigation of the *trw* gene cluster and it very nicely shows that genes can be disabled without expression of others being disrupted.

Since these complementation studies have already been carried out, we thought to address the reviewer #1's comment in a different, and we would like to believe, perhaps a better way. Indeed, to address directly the issue raised by the reviewer, for each deletion constructs for which an effect on complex formation was observed (*ΔtrwM/virB3*, *ΔtrwK/virB4*, *ΔtrwI/virB6*, and *ΔtrwG/virB8*), we introduced sequences encoding FLAG tags to the 5'- or 3'-end of each component gene that we observed were missing after pull-down and monitored expression of the corresponding proteins. Since, in the *ΔtrwM/virB3* deletion, we observed the loss of TrwK/VirB4, TrwJ/VirB5, TrwI/VirB6 and TrwG/VirB8, four derivatives of this *ΔtrwM/virB3* construct were generated in order to express variants of these four proteins containing a FLAG tag at either their N- or C-terminus (Table S1). Similarly, the *ΔtrwK/virB4* deletion results in the loss of the TrwM/VirB3, TrwJ/VirB5, TrwI/VirB6 and TrwG/VirB8 proteins: thus, we generated four further variants of the *ΔtrwK/virB4* construct where these proteins were FLAG-tagged individually (Table S1). For the *ΔtrwI/virB6* deletion, we observed the loss of TrwJ/VirB5 and consequently a FLAG sequence was added to TrwJ/VirB5 and its expression checked (Table S1). Finally, deletion of TrwG/VirB8 resulted in the destabilisation of TrwM/VirB3, TrwK/VirB4, TrwJ/VirB5, and TrwI/VirB6 and therefore four additional constructs of this deletion mutant were generated to include a FLAG sequence in each of these proteins, one at a time (Table S1). To compare expression levels of each of the FLAG-tagged proteins within deletion mutants with the level of expression in the non-deleted wild-type pBADM11-*trwN/virB1-trwE/virB10_{Strept}-trwD/virB11_{His}-trwB/virD4* strain, FLAG sequences were also inserted in this construct at the same position as in the deletion mutants, one at a time. After induction, expression was monitored by Western blot analysis using an anti-FLAG antibody. As can be seen in the new Figure S1B, all proteins are expressed. Lower expression compared to wild-type is however observed for TrwM/VirB3 in the *ΔtrwK/virB4* construct and for TrwI/VirB6 in the *ΔtrwM/virB3* and *ΔtrwK/virB4* constructs. Could this be due to a polar effect on expression? Since, in the *trw* operon structure, the order of genes is *trwL,M,K,J,I/virB2,3,4,5,6*, a polar effect in the *ΔtrwM/virB3* and *ΔtrwK/virB4* constructs would affect expression of *trwJ/virB5*; yet expression of this protein is unaffected in these constructs (new Figure S1B). Therefore, lower production of VirB3 and VirB6 in these constructs might be due to an effect of the removed protein on the stability of the TrwM/VirB3 or TrwI/VirB6 proteins. Nevertheless, TrwM/VirB3 and TrwI/VirB6 are produced in the *ΔtrwM/virB3* and *ΔtrwK/virB4* constructs and, although the level of expression is lower than in the corresponding wild-type constructs, it is similar to that observed for TrwJ/VirB5 in all constructs including wild-type. We also checked for expression of *trwB/virD4* in all deletion clusters, and showed that each produced similar quantities of TrwB/VirD4 (new Figure S1A). We therefore conclude that the loss of particular T4SS components during pull-down that we observe when some T4SS components are removed is not due to the lack of expression of these particular components.

After having addressed the editor's comments, we now would like to provide a point-by-point description of how we have addressed the reviewers' suggestions.

Referee #1:

Major issues:

1. Deficiencies need to be rectified with the 3-plasmid system.

A) Stated overproduction. The authors present no evidence that the Trw were overproduced when expressed from inducible promoters. Since only epitope-tagged proteins were evaluated, not the native proteins, no comparisons could be made among steady-state levels of epitope-tagged proteins produced from the inducible promoters and native proteins produced from the native promoter. The Llosa lab has antibodies to many of the Trw proteins, why not request those antibodies for direct comparisons?

B) Stated novelty: The authors overstate the novelty of their 3-plasmid system. In fact, such systems exist in nature and have also been previously engineered by other laboratories. For example, in nature genes encoding the T4SSs are generally expressed separately from coupling proteins - either on the same plasmid or on separate plasmids. See papers by Llosa et al. documenting transmissibility of plasmids carrying the R388 mob genes from the T4SS expressed from a separate plasmid. The capacity of T4SSs to mobilize oriT-containing plasmids or chromosomal fragments is also well established - see work from Meyer's lab as an example (there are many others). The authors should tamp down their claim of novelty and instead cite this large literature demonstrating the modular nature of T4SSs.

C) Why is the cloning of the trw locus behind inducible promoters, and then the subsequent demonstration of functionality considered 'reconstitution'? Basically, a locus was placed under the control of a regulatable promoter, this isn't reconstitution, it's simply cloning.

Response to A), B) and C)

We apologise for having over-emphasize these aspects of the work. As a result of the reviewer's very pertinent comments, we have completely rewritten this part of the manuscript. i- Although our design is satisfactory in terms of producing sufficient quantities of material for all aspects of our investigation, we no longer claim that it is over-produced. ii- We apologise for having overstated the novelty aspects of our design; we have removed this claim and we acknowledge the contribution of Llosa and colleagues. iii- We have also removed all mention of "reconstitution" as our 3-plasmid system is indeed "cloning".

2. Effects of individual trw gene deletion on machine assembly.

A) The above points are also important because many tra loci are not expressed from a single upstream promoter and to this reviewer's knowledge there is no demonstration that all of the trw genes are expressed from a single promoter. Also, there are many instances in which deletion of a gene disrupts expression of upstream or downstream genes (e.g., due to translational coupling), but not other genes in the locus. Since the authors do not have antibodies to carefully address this question, minimally, complementation studies (by expression of the deleted gene from another plasmid) must be carried out to show that the deletion is directly responsible for altered protein content of the isolated subassemblies. Clearly, at this time, it is insufficient to show levels of VirD4(TrwB) expressed from the most distal gene in this locus, as its expression could be under the control of a constitutive internal promoter.

Response

Please see response to the editor's request #4. We feel we have addressed the issue raised here by the reviewer in a more thorough way than what the reviewer here suggests: we indeed have directly monitored the production of the components that are missing in our pull-downs for each deletion mutant where we observe an effect on complex formation. As reviewer 1 very justifiably argues, if protein B is missing when protein A is removed, it could be that it is because the deletion of the

protein A-encoding gene leads to protein B not being produced. Therefore, we have FLAG-tagged protein B and checked that it is indeed produced using Western blot analysis. As is explained in our response above, all proteins are produced in all deletions and therefore the results of our pull-down are valid: if a protein is missing in our pull-down, it is not because it hasn't been produced.

B) Some of the results presented in Figure 2 were already published by the Llosa lab (Llarea et al. 2013, Plasmid). Surprisingly this paper was not even cited - the paper should be cited and the present and previous findings compared.

Response

We have apologised above for not citing this paper. We now do and we have a short (due to space constraints) commentary on its main results. It is however important to realise that Larrea et al (2013) have very different goals than ours: they are looking at protein stability in the absence of various components as assessed by Western analysis of cell lysates. We have an entirely different goal and approach, which is to assess the importance of each component in T4SS_{3-10+D4} complex formation as assessed by pull-down and purification of the resulting complexes when single components are deleted. Nevertheless, now that we provide quantifications of expression for each component that are missing in the pull-down complexes upon deletion of other components, we can compare the two studies and, to our surprise, they are somewhat different. One striking difference is that, in our hand, we see expression of TrwJ/VirB5 in all deletion mutants. We have added a short sentence to point to these differences.

C) Larrea et al. also reported that several trw mutations affect levels of TrwD(B11); the authors should attempt to reconcile why this protein is not copurified with their complexes.

Response

We are somewhat confused by this comment. As mentioned in our previous version of the manuscript, TrwD/VirB11 does not co-purify with either the T4SS₃₋₁₀ and the T4SS_{3-10+D4} complexes. We suspect that this is because TrwD/VirB11 being a soluble protein superficially tethered to the membrane, its interaction with T4SS₃₋₁₀ and the T4SS_{3-10+D4} is perturbed by detergents and therefore it falls off during detergent-solubilisation of the complex during purification. As a result, we cannot explore its interaction with other proteins and therefore cannot comment on Larrea et al. results on this particular issue.

3. The localization patterns observed for gfp- and FLAG-tagged VirB8(trwG) do seem to be similar to those seen by Zambryski for the Agrobacterium proteins. However, this is the weakest section of the manuscript for the following:

A) Data obtained with GFP-tagged protein cannot be relied on because the GFP tag disrupts protein function. Data obtained with the FLAG-tagged protein seem ok.

B) The authors do not show the localization pattern of GFP-B8 in the absence of the other machine subunits (although it is stated to be uniform - needs to be shown).

C) The colocalization of this protein with the T4SS would be more convincing if the spatial localization of more than one Trw protein were analyzed.

D) The comparison with Agrobacterium proteins is interesting, but in those studies the authors also almost exclusively analyzed attenuated or nonfunctional gfp-tagged proteins that were expressed from a plasmid other than that encoding the rest of the Vir proteins. It isn't clear why the authors think that the Agrobacterium system represents an 'endogenous' system while theirs is a 'reconstituted' system. Also, in fact the Agrobacterium system is a conjugation system, so it is wrong to state that the present study is the first to show a supposedly helical pattern for a conjugation system. It is interesting that both systems seem to show similar patterns, but that is the extent of the conclusions allowed by the available data.

E) The use of higher resolution microscopy (TIRF, SIM, cryoEM) has raised questions of whether even well-characterized proteins such as MreB form helical arrays or instead assemble as discrete

disconnected patches. In view of this ongoing debate and based on the data presented, it is premature to conclude that this protein or the T4SS forms helical arrays.

Response to A), B), C), D) and E)

Please see our response to the editor's request #1. This part of the paper has been criticised by all reviewers and, as reviewer #1 rightly points out, it does not constitute the most compelling and interesting part of the manuscript. We have therefore taken up the editor's suggestion that we removed this part entirely.

Minor issues:

4. Pg. 5. Top. VirB4 and VirB11 are also required for substrate translocation - not just pilus biogenesis, as stated.

Response

This has been corrected.

5. Pg. 5. Top. Considerable biochemical work was done on the VirB4 ATPases by Eric Lanka and others. These studies provided the first evidence for different oligomeric forms of these ATPases, and should be cited.

Response

We have added these references

6. Pg. 7. Bottom. *trwA* and *ihf*. If these are references to bacterial proteins, they should have the first letter capitalized and not be italicized.

Response

We have corrected

7. Pg. 11. Middle. The last sentence of this section makes no sense. The 'endogenous' system that needs to be compared is the R388 T4SS produced by R388 vs the inducible expression systems (which are not 'reconstituted' systems, but simply *tra* operons cloned behind inducible promoters). Overproduction of the *tra* proteins from the inducible promoters might lead to aberrant localization.

Response

We removed the fluorescence microscopy section of the paper and therefore, this passage is no longer included.

8. Pg. 11. What does "successfully integrate in the cell envelope" mean? This system promotes conjugative DNA transfer, so it must be successfully integrating into the cell envelope. The fluorescence microscopy data do not allow for a discrimination between 'successful' and 'nonsuccessful' integration.

Response

This paragraph is part of the fluorescence microscopy section that has been deleted as suggested by the editor and all reviewers. So, it is no longer a part of our manuscript.

9. Pg. 12. For the stoichiometric analyses, the gels should be shown in the supplementary files.

10. Pg. 14. Middle. Complementation studies need to be carried out to confirm that the deletions did not have polar effects on expression of genes either upstream or downstream of the mutation. This is particularly important for mutations that appear to have disrupted machine production, e.g., *virB3*, *virB4*, *virB8* deletions. Expression *virB8* from another plasmid should both restore functionality of the T4SS and levels of *VirB4*. Similarly, expression of *virB6* should restore *VirB5* levels in the isolated machine. These studies are necessary to eliminate possible artifacts resulting from the disruption of adjacent gene expression through various mechanisms.

Response

Please see response to the editor's request #3. The gel is now shown. Also, concerning potential polar effects, Please see response to the editor's request #4. We believe that, with our new results, we have addressed these issues and that all artifacts resulting from the disruption of adjacent gene expression have been addressed and proven to be eliminated.

11. In view of the demonstrated interaction between TrwB and TrwE and its proposed importance, why not show that a trwE deletion mutation abolishes recovery of TrwB from the isolated machines?

Response

We now present an entire series of experiments to address this issue. Indeed, as the reviewer points out, our investigation of complex formation of TrwB/VirD4 with the T4SS₃₋₁₀ complex points to the importance of the interaction between TrwB/VirD4 with the core TrwH/VirB7-TrwF/VirB9-TrwE/VirB10 complex. Interaction between TrwB/VirD4 and TrwE/VirB10 has been previously observed, and, therefore, we set out to confirm it biochemically in the context of the entire T4SS_{3-10+D4} complex. Within the pBADM11_ *trwN/virB1-trwE/virB10*_{Strept_}*trwD/virB11*_{His}*trwB/virD4* cluster, we deleted sequences encoding either the entire cytoplasmic tail of TrwE/VirB10 (residues 1-42; referred to as T4SS_{3-10+D4:TrwE42}) or both the cytoplasmic and trans-membrane segment of this proteins (residues 1-64; referred to as T4SS_{3-10+D4:TrwE64}). After induction of expression, the T4SS_{3-10+D4} complex and variants were purified and analysed by SDS-PAGE (new Figure 2). Expression of the T4SS_{3-10+D4:TrwE42} resulted in lower amounts of TrwB/VirD4 bound to the complex indicating that indeed the cytoplasmic N-terminal region of TrwE/VirB10 is important for interaction with TrwB/VirD4. Removal of the trans-membrane segment of TrwE/VirB10 further destabilises the interaction with the coupling protein, confirming that some of the interactions between TrwE/VirB10 and TrwB/VirD4 involve the trans-membrane segment of both proteins.

12. Shifting between the Trw and Vir nomenclature confusing. This work focuses on the R388 system, so all references should be to the Trw proteins. A table could be included in the Suppl identifying the VirB homologs.

Response

We now use both nomenclatures throughout the manuscript. This has the advantage of not having to consult a table whenever the two names are needed. This notation is now quite common in the field (see Larrea et al. (2013), Fronzes et al. (2009) or Low et al. (2014)).

Referee #2:

Major

1. P6 third line from the bottom- do the authors actually '...determine the stoichiometry of all components within this complex' in this manuscript?

Response

Yes, we have. Indeed, using Cys-labelling, all components containing Cys residues are labelled and therefore all these components' stoichiometry can be measured. We found that the stoichiometry for the TrwM/VirB3-TrwE/VirB10 proteins in the T4SS₃₋₁₀ and T4SS_{3-10+D4} complexes are the same. For this revised version of the manuscript, we re-assessed the stoichiometry for TrwB/VirD4 in the T4SS_{3-10+D4} complex: please see details in our response to editor's request #3 above. In the text, this point is addressed on Page 11.

2. P10 and Fig 1 It would be useful to include the data on the distribution of fluorescence from eGFP-VirB8 alone, for comparison, to clarify the point made about localization of the reconstituted conjugative T4S systems into discrete foci.

Response

Please see our response to editor's request #1. This part of the paper has been removed.

3. P15 Line 9 What is the evidence that the barrel-like densities have 3-fold symmetry?

Response

This has been amply discussed for the T4SS₃₋₁₀ complex by Low et al. (2014). We observe the same in the T4SS_{3-10+D4} complex. To illustrate this fact, we now provide a bottom view in Figure 4 that clearly shows three patches of density for each barrel.

4. P15 The evidence for the location of VirD4 in the 3D volume is derived from differences in projections between T4SS_{3-10+D4} and T4SS₃₋₁₀ in Fig 2. Was the 3D difference map also calculated? It would be clearer and more convincing to present that, in Fig 4A, for example. It should also be acknowledged somewhere that 3D volumes derived from negative stain are subject to flattening and distortion.

Response

We apologise for having been insufficiently clear here, but the location of VirD4 in the 3D volume was NOT ONLY derived from differences in projections between the two complexes, BUT ALSO derived from the difference in 3D volumes between the two complexes (See old Figure 4). However, it is true that we did not show any 3D difference map and this has now been remedied (See new Figure S2A). We also now acknowledge the flattening and distortion that may be caused by negative stain.

5. P18 Line 9 Can the authors be confident that the electron density only accommodates two VirD4 monomers and not, for example, one or three? Given previous comments on the limitations of negative stain-derived volumes and stoichiometry estimates, a suitable caveat should be applied to this statement.

Response

The reviewer is right and we have added a suitable caveat on Page 17.

6. Presumably the T4SS_{3-10+D4} 3D volume will be deposited in the EMDB?

Response

We have now deposited the 3D volume and have included the EMDB entry code (EMD-3585)

Minor issues:

1. P6-7 The end of the Introduction contains an unnecessarily long summary of the main conclusions of the work; this could be pruned.

Response

We have pruned the introduction, notably by removing the claims of over-expression, reconstitution and 3-plasmid design novelty as requested by reviewer 1 and also the removal of the fluorescence microscopy results.

2. P7 7th line from top '..and located at..'

Response

We have corrected

3. P9 8th line from top '..a 5-fold decrease...' According to Fig 1A, the mating efficiency of pAsk_{virB1-11} is 0.4 and pAsk_{virB1_B8eGFP-virB9-11} is 0.02, which is a 20-fold difference?

Response

We apologise for the mistake. However, this part of the manuscript has now been entirely removed as we now longer present the fluorescence studies.

4. P10 Line 15 should 'pAsk_{virB1-7_eGFPvirB8-11}' be 'pAsk_{virB1_B8eGFP-virB9-11}'?

Response

We apologise for the mistake. However, this part of the manuscript has now been entirely removed as we now longer present the fluorescence studies.

5. Figure 2: mass markers are needed for the gel in Fig 2B.

Response

Molecular weight markers are now included (now Figure 1C).

Referee #3:

In this manuscript Redzej and coauthors describe the negative stain electron microscopy structure of a complex comprising the TrwE-K, TrwM and TrwB proteins of the R388 conjugative Type IV secretion system. Compared to the previous structure of the TrwE-K-TrwM complex published by the same group in 2014, the TrwB coupling protein has now been added to the picture.

Comments.

- I do not understand the rationale to use the VirB nomenclature. The VirB proteins are the homologs in the Agrobacterium system, and it remains possible that the VirB system is structurally different. The Trw nomenclature (used in all the publications on this system since 1990!) should be used.

Response

We apologise for not using the Trw nomenclature. We have now corrected and it is used throughout.

- Mating experiments. A negative control should be provided (pAsk and pBADM11 plasmids expressing TrwD-N with the TrwAC relaxosome plasmid).

Response

We now have the requested negative control (pBADM11 plasmids expressing TrwD-N with the TrwAC relaxosome plasmid).

- Some of the data presented are of poor quality, specifically the fluorescence microscopy experiments showed in Fig. 1 which have a very low resolution. Note that the presentation of these fluorescence experiments is sloppy as no statistical analysis is provided nor scale bars.

Response

Please see our response to editor's request #1. This part of the paper has now been removed.

- The authors claim that the machinery locates on a helical pattern but the deconvolution images are not convincing - additional statistical analyses should be performed. The conclusion is also overstated as the authors imaged GFP-TrwG and there is no evidence that the discrete foci corresponds to assembled machinery rather than TrwG hetero-oligomeric complexes. In addition the R388 conjugative system is overproduced and the localisation observed may not reflect the real situation.

Response

Please see our response to editor's request #1. This part of the paper has now been removed.

- Fig. 2B. TrwB associates with the different complexes lacking one subunit. However, TrwE is not tested as the purification tag is fused to this subunit. The authors should test a construct in which the TrwE transmembrane helix has been deleted.

Response

Please see our response to comment 11 or reviewer 1 and repeated here for the reviewer's convenience:

We now present an entire series of experiments to address this issue. Indeed, as the reviewer points out, our investigation of complex formation of TrwB/VirD4 with the T4SS₃₋₁₀ complex points to the importance of the interaction between TrwB/VirD4 with the core TrwH/VirB7-TrwF/VirB9-TrwE/VirB10 complex. Interaction between TrwB/VirD4 and TrwE/VirB10 has been previously observed, and, therefore, we set out to confirm it biochemically in the context of the entire T4SS_{3-10+D4} complex. Within the pBADM11_*trwN/virB1-trwE/virB10_{Strep}-trwD/virB11_{His}-trwB/virD4* cluster, we deleted sequences encoding either the entire cytoplasmic tail of TrwE/VirB10 (residues

1-42; referred to as T4SS_{3-10+D4:TrwE-42}) or both the cytoplasmic and trans-membrane segment of this proteins (residues 1 -64; referred to as T4SS_{3-10+D4:TrwE-64}). After induction of expression, the T4SS_{3-10+D4} complex and variants were purified and analysed by SDS-PAGE (new Figure 2). Expression of the T4SS_{3-10+D4:TrwE-42} resulted in lower amounts of TrwB/VirD4 bound to the complex indicating that indeed the cytoplasmic N-terminal region of TrwE/VirB10 is important for interaction with TrwB/VirD4. Removal of the trans-membrane segment of TrwE/VirB10 further destabilises the interaction with the coupling protein, confirming that some of the interactions between TrwE/VirB10 and TrwB/VirD4 involve the trans-membrane segment of both proteins.

- Fig. 2B. The authors claim that TrwJ is no longer present in the complex in the absence of TrwI. However, a SDS-PAGE or western blot analysis of the cell lysate should be provided. Is TrwJ produced?

Response

We now provide proof that in the absence of TrwI/VirB6, TrwJ/VirB5 is produced. Indeed, we have added a FLAG-tag in TrwJ/VirB5 in the the *AtrwI/virB6* deletion mutant and, using Western blot analysis, we show that TrwJ/VirB5 is produced in similar quantities as in the wild-type construct.

- The data regarding the Alexafluor633 labeling to report the stoichiometry should be shown. The standard deviation obtained (+/- 60%) is rather huge. The stoichiometry of TrwB in the complex should be addressed by an independent approach such as native mass spectrometry.

Response

Please see our response to editor's request #1. There is no native mass spectrometry method or instrument that can deal with such a large complex. We now show the stoichiometry results.

- Fig. 3C. It is not clear whether the TrwE-K/TrwM complex has been subjected to GraFix too.

Response

The T4SS₃₋₁₀ (TrwE-K/TrwM) complex has not been subjected to GraFix. The only reason we used GraFix for the T4SS_{3-10+D4} complex is because of the tendency of TrwB/VirD4 to dissociate from the complex. For the T4SS₃₋₁₀ complex, GraFix was not necessary because we didn't observe any loss of components during and after purification. One could argue whether it is valid to compare the structure of a complex which has been submitted to GraFix with another one that has not: in our view, it is perfectly legitimate and valid as GraFix is a very mild crosslinking method unlikely to introduce any artefact. Moreover, we now present a series of extensive immuno-labelling studies that confirm by entirely different means the location of TrwB/VirD4 (see response to editor's request #2, repeated as a response to the next comment below for convenience). Finally, our XL-MS data is also consistent with the observed location of TrwB/VirD4 in the complex. Overall, there is no doubt as to the location of TrwB/VirD4 in the T4SS_{3-10+D4} complex.

- Whereas the location of TrwB is defined based on comparison with TrwBless complexes, a definitive demonstration is lacking such as immunogold labeling or purification of TrwE-K/M complex with TrwB-GFP.

Response

Please see response to editor's request #2 and repeated here for the reviewer's convenience: We now provide further support for the location of TrwB/VirD4. Using immuno-labelling followed by electron microscopy, we were indeed able to confirm independently the location of TrwB/VirD4. Immuno-labelling by NS-EM provides a means by which the location of a particular component of a large complex can be located within the electron density of that complex. In brief, an antibody targeting a particular component of a complex is reacted to the complex; the resulting antibody-bound complex is then imaged by NS-EM and class averages of antibody-bound complex particles is compared to class averages of non-reacted complexes. Extra electron density corresponding to the antibody should be observed where the antibody has bound and this electron density should be located within close proximity of the component to which the antibody has bound. To immuno-label TrwB/VirD4 within the larger T4SS_{3-10+D4} complex, we generated a variant of the *AtrwD/virB11* construct with a FLAG tag-encoding sequence introduced at residue Thr236 of TrwB/VirD4. The 236 position in the protein structure is shown as a red sphere in the new Figure

5A of this revised version of the manuscript. This region belongs to the cytoplasmic domain of TrwB/VirD4 and is at the opposite end of the structure relative to the N-terminus which is known to be inserted in the inner membrane. Therefore, a FLAG tag inserted in this region should be easily accessible for binding by anti-FLAG antibodies and, if our localisation of TrwB/VirD4 as seen in the NS-EM structure of the T4SS_{3-10+D4} complex (Figure 4) is correct, the bound antibodies should be observed on the side or at the bottom of the IMC in the NS-EM class averages of the antibody-bound complex.

The FLAG-tagged complex, termed “T4SS_{3-10+D4FLAG}” was purified and reacted to anti-FLAG antibodies. The antibody-bound complex was further purified, applied to a grid and stained using negative stain as described in Materials and Methods. After data collection, class averages were obtained: these clearly show extra-density for the FLAG-antibody in close proximity of TrwB/VirD4, either on the side or near the bottom (new Figure 5B). Difference density (new Figure S3A, right panels) between class averages of the T4SS_{3-10+D4} (new Figure S3A, left panels) and antibody-bound T4SS_{3-10+D4FLAG} (Figure S3A, middle panel) clearly illustrates the presence of the extra density generated by antibody-binding. Thus, the additional density observed by comparing the T4SS₃₋₁₀ and T4SS_{3-10+D4} (new Figure 4) can be indeed safely attributed to TrwB/VirD4.

We would like to thank the reviewers for their immense contribution to this manuscript. Their comments and suggestions have resulted in the vastly-improved paper which we believe is now ready for publication in EMBO Journal.

2nd Editorial Decision

03 August 2017

Thank you for submitting a revised version of your manuscript. I apologise for the delay in communicating the decision due to the delay in receiving referee reports and the busy conference schedule at this time of the year. We have now received reports from all original reviewers. As you will see, referees #1 and #2 find that their main concerns have now been addressed, while referee #3 requests expansion of the structure and interaction network analysis. In a further consultation reviewers #1 and #2 indicated that these experiments are beyond the scope of the study and therefore do not have to be addressed for the acceptance here.

I would like to ask you to address the few remaining mainly editorial issues in the final version of the manuscript before I can send a formal acceptance letter:

1. Please address the remaining comments of referees #1 and #2.
2. Please modify the title according to the suggestions of reviewer #3 to focus on presentation of a new T4SS structure with the docked VirD4 coupling protein.
3. Figure 2 (the right and middle panels) and Figure S1 appear highly contrasted, please replace with less modified images.
4. Please add the molecular weight markers in Figure 1A and 1B.
5. Please update the references according to The EMBO Journal style (where there are more than 20 authors on a paper, the first 20 should be listed, followed by 'et al.').
6. Please convert supplemental images into Expanded View figures, which are then displayed together with the corresponding main figure in the online version of the manuscript, thus increasing the accessibility of supplemental data. Please see our author guidelines on details about the content, purpose and preparation of Expanded View material (<http://emboj.embopress.org/authorguide#expandedview>).
7. Please assemble the Appendix tables into a single pdf with a table of contents as described in our Author Guidelines: <http://emboj.embopress.org/authorguide#expandedview>.
8. If you have not yet done so, please add ORCID ID for Elena Orlova, as we require this for all corresponding authors. In order to link your ORCID iD to your account in our manuscript tracking system, please do the following:
 - Click the 'Modify Profile' link at the bottom of your homepage in our system.
 - On the next page you will see a box halfway down the page titled ORCID*. Below this box is red text reading 'To Register/Link to ORCID, click here'. Please follow that link: you will be taken to ORCID where you can log in to your account (or create an account if you don't have one)
 - You will then be asked to authorise Wiley to access your ORCID information. Once you have approved the linking, you will be brought back to our manuscript system. We regret that we cannot

do this linking on your behalf for security reasons.

9. We generally encourage the publication of source data, particularly for electrophoretic gels and blots, with the aim of making primary data more accessible and transparent to the reader. We would need 1 file per figure (which can be a composite of source data from several panels) in jpg, gif or PDF format, uploaded as "Source data files". The gels should be labelled with the appropriate figure/panel number, and should have molecular weight markers; further annotation would clearly be useful but is not essential. These files will be published online with the article as a supplementary "Source Data".

Finally, papers published in The EMBO Journal include a 'Synopsis' to further enhance discoverability. Synopses are displayed on the html version of the paper and are freely accessible to all readers. The synopsis includes a short introductory paragraph as well as 2-5 one-sentence bullet points that summarise the paper and are provided by the authors. Please send us your suggestions for bullet points and a synopsis image. This image should provide a rapid overview of the question addressed in the study, but still needs to be kept fairly modest, since the image size cannot exceed 550x400 pixels.

Please let me know if you have any further questions regarding this final revision. You can use the link below to upload the revised version.

REFEREE REPORTS

Referee #1:

In this revised manuscript, the authors have carefully addressed all of this reviewer's previous concerns. The manuscript reports important new information pertaining to the spatial location of the substrate receptor for the type IV secretion machine that this group has been working on for many years. The work is much better integrated than the original version, and additional data have been included to shore up the major conclusions. Most significantly, the authors present strong evidence for the association of the VirD4 subunit at the base of the T4SS machine, sandwiched as a homodimer between the two VirB4 hexamers. They confirm the identity of the extra electron density seen in the T4SS3-10+D4 structure as compared with the T4SS3-10 structure alone by use of antibody labeling of a FLAG-tagged D4 variant, and they also present evidence for the importance of the N-terminal region of the VirB10 subunit for proper positioning of the D4 dimer, in agreement with previous evidence for D4-B10 interactions. The authors also have gone to great lengths to confirm production (stability) proteins in various mutant backgrounds as a prerequisite for interpreting structures recovered from strains lacking individual machine subunits. The work is really a tour de force and a wonderful contribution to the field. It greatly advances a structural definition of a paradigmatic T4SS, and the authors have now nicely integrated their findings with the previous literature. The new structural information also raises intriguing mechanistic questions for future investigations pertaining to the coordination of functions of the 3 ATPases of T4SSs in docking and then driving the translocation of substrates across the bacterial cell envelope.

This reviewer has only a few comments to be addressed in the interest in improving an already very solid piece of work.

1. Nomenclature: I agree that it is important to present both the Trw and VirB names since the former are the actual names and the latter the unifying nomenclature for T4SSs. But, as presented, some constructs are awfully long and difficult to interpret. I wonder if the use of subscripts for the VirB names would help simplify things, e.g., trwEB₁₀ and TrwEB₁₀ for gene and protein, respectively (these subscripts didn't get incorporated when I copy-pasted the word file into this review section, but you know what I mean). In relation to this, the Table S2, still reports the interactions between the VirD/B subunits as opposed to the Trw subunits.

2. Genes are expressed and proteins are produced or synthesized. In the interest of accuracy - and often to clarify meaning - the authors should adhere to this verbiage. Additionally, protein production should not be equated with steady-state levels or abundance, as a produced protein might not be detected due to instability. This is relevant when referring to steady-state levels of subunits in various deletion mutant strains.

3. The crosslinking data nicely complement the structural findings, but some of the reported crosslinks are confusing in the context of those findings. For example, the extreme C terminus of TrwBD4 is reported to interact with an N proximal domain of TrwKB4. Similarly, the C-terminus of TrwBD4 interacts with the extreme N terminus of TrwEB10. Neither of these crosslinks seem to square with the structural data showing that ATPase multimers align such that their N-terminal regions are juxtaposed with the inner membrane, or with results of previous studies suggesting that the TrwBD4 and TrwEB10 interact via their N termini. Explanation?

Referee #2:

The revised manuscript is much improved, with new data on the location of TrwB/VirD4, stoichiometry measurements and verification of expression through the use of FLAG tags. One minor point, the labelling of mass standards on the gels is still not clear; are the standards in Fig 1C the same as Fig 1A, for example? Similar details should be supplied for Fig 2 and Fig S1.

Referee #3:

In this revised manuscript Redzej et al provide additional data compared to the initial submission. Specifically, they have added controls for mating experiments, have performed the cys-labeling stoichiometry with higher standards, and have provided better evidence for localization of TrwB/VirD4 using a bound antibody. The authors have decided to delete all the fluorescence microscopy data rather than expanding their findings to better standards. The consequences of these modifications are that the manuscript is now structurally-only oriented and presents limited new information (the position of TrwB in the complex). For a manuscript that only reports the structure of the complex in absence of in vivo data, it is now admitted that vitrified specimens should be imaged by cryo-electron microscopy. In addition, the "interaction network" part needs to be substantiated. While cross-linking data are provided to identify contacts between TrwB and the R388 conjugation machinery, these contacts are not validated using site-directed mutagenesis and co-purification assays, and their relevance in DNA transfer is not addressed. The title is also misleading: the manuscript does not provide data on how VirD4 "energizes" the VirB machinery.

2nd Revision - authors' response

09 August 2017

Please find enclosed a second revised version of our manuscript entitled "**Energizing the T4S system: structure of a VirD4 coupling protein bound to a VirB machinery**" by Adam Redzej, Sarah Connery, Marta Ukleja, Martina Trokter, Catarina Felisberto-Rodrigues, Adam Cryar, Konstantinos Thalassinou, Richard D. Hayward, Elena V. Orlova and Gabriel Waksman.

You and the reviewers have made a number of excellent suggestions that we have incorporated the following way:

1. Please address the remaining comments of referees #1 and #2.

Response: We list below the comments by referees 1 and 2 and provide details as to how we have addressed them.

Referee #1:

1. Nomenclature: I agree that it is important to present both the Trw and VirB names since the former are the actual names and the latter the unifying nomenclature for T4SSs. But, as presented, some constructs are awfully long and difficult to interpret. I wonder if the use of subscripts for the VirB names would help simplify things, e.g., trwEB_{10} and TrwEB_{10} for gene and protein, respectively (these subscripts didn't get incorporated when I copy-pasted the word file into this review section, but you know what I mean). In relation to this, the Table S2, still reports the interactions between the VirD/B subunits as opposed to the Trw subunits.

Response: We have followed this recommendation throughout.

2. Genes are expressed and proteins are produced or synthesized. In the interest of accuracy - and often to clarify meaning - the authors should adhere to this verbiage. Additionally, protein production should not be equated with steady-state levels or abundance, as a produced protein might not be detected due to instability. This is relevant when referring to steady-state levels of subunits in various deletion mutant strains.

Response: We have corrected.

3. The crosslinking data nicely complement the structural findings, but some of the reported crosslinks are confusing in the context of those findings. For example, the extreme C terminus of TrwBD4 is reported to interact with an N proximal domain of TrwKB4. Similarly, the C-terminus of TrwBD4 interacts with the extreme N terminus of TrwEB10. Neither of these crosslinks seem to square with the structural data showing that ATPase multimers align such that their N-terminal regions are juxtaposed with the inner membrane, or with results of previous studies suggesting that the TrwBD4 and TrwEB10 interact via their N termini. Explanation?

Response: We have added on page 19: “The observed crosslinks between $\text{TrwB}_{\text{VirD4}}$ and $\text{TrwE}_{\text{VirB10}}$ or $\text{TrwB}_{\text{VirD4}}$ and $\text{TrwK}_{\text{VirB4}}$ are consistent with structural knowledge since the residues involved in the crosslinks are all likely to locate near the cytosolic side of the inner membrane”.

Referee #2:

1. One minor point, the labelling of mass standards on the gels is still not clear; are the standards in Fig 1C the same as Fig 1A, for example? Similar details should be supplied for Fig 2 and Fig S1

Response: The labelling of mass standards in Fig 1A and 1C are the same. We have now added a note to that effect in the figure legend. We have added the labelling in Fig 2 and Fig S1.

2. Please modify the title according to the suggestions of reviewer #3 to focus on presentation of a new T4SS structure with the docked VirD4 coupling protein.

Response: We have now changed the title to “Structure of a VirD4 coupling protein bound to a VirB type IV secretion machinery”.

3. Figure 2 (the right and middle panels) and Figure S1 appear highly contrasted, please replace with less modified images.

Response: These are how the original gels look like (see new source data file). So we are not quite sure how to implement the suggested modification.

4. Please add the molecular weight markers in Figure 1A and 1B.

Response: As requested by Reviewer 2, we now have added labelling for all molecular weight markers except for Fig 1A and 1C where the markers are the same and a note to that effect has been added in the figure legend.

5. Please update the references according to The EMBO Journal style (where there are more than 20 authors on a paper, the first 20 should be listed, followed by 'et al.').

Response: We have now corrected.

6. Please convert supplemental images into Expanded View figures, which are then displayed together with the corresponding main figure in the online version of the manuscript, thus increasing the accessibility of supplemental data. Please see our author guidelines on details about the content, purpose and preparation of Expanded View material (<http://emboj.embopress.org/authorguide#expandedview>).

Response: We have now corrected.

7. Please assemble the Appendix tables into a single pdf with a table of contents as described in our Author Guidelines: <http://emboj.embopress.org/authorguide#expandedview>.

Response: We have done so.

8. If you have not yet done so, please add ORCID ID for Elena Orlova, as we require this for all corresponding authors.

Response: This has also been done.

9. We generally encourage the publication of source data, particularly for electrophoretic gels and blots, with the aim of making primary data more accessible and transparent to the reader. We would need 1 file per figure (which can be a composite of source data from several panels) in jpg, gif or PDF format, uploaded as "Source data files". The gels should be labelled with the appropriate figure/panel number, and should have molecular weight markers; further annotation would clearly be useful but is not essential. These files will be published online with the article as a supplementary "Source Data".

Response: We are now including a source data file.

We would like to thank you and the reviewers for their immense contribution to this manuscript. Their comments and suggestions have resulted in the vastly-improved paper which we believe is now ready for publication in EMBO Journal.

3rd Editorial Decision

22 August 2017

Thanks very much for resolving the few remaining issues in the manuscript. I am now pleased to inform you that your manuscript has been accepted for publication in the EMBO Journal. Congratulations on a nice study!

Corresponding Author Name: Gabriel Waksman
 Journal Submitted to: EMBO Journal
 Manuscript Number: EMBOJ-2017-96629